# Comprehensive analysis of lectin-glycan interactions reveals determinants of lectin specificity

**Daniel E. Mattox**[1], **Chris Bailey-Kellogg**[1,2]*

**1** Program in Quantitative Biomedical Sciences, Geisel School of Medicine at Dartmouth College, Hanover, New Hampshire, United States of America, **2** Department of Computer Science, Dartmouth College, Hanover, New Hampshire, United States of America

* cbk@cs.dartmouth.edu

**Data Availability Statement:** The GitHub repository https://github.com/demattox/lec_gly_binding contains the scripts used to perform the statistical and predictive analyses presented here, along with the combined UniLectin3D information

## Abstract

Lectin-glycan interactions facilitate inter- and intracellular communication in many processes including protein trafficking, host-pathogen recognition, and tumorigenesis promotion. Specific recognition of glycans by lectins is also the basis for a wide range of applications in areas including glycobiology research, cancer screening, and antiviral therapeutics. To provide a better understanding of the determinants of lectin-glycan interaction specificity and support such applications, this study comprehensively investigates specificity-conferring features of all available lectin-glycan complex structures. Systematic characterization, comparison, and predictive modeling of a set of 221 complementary physicochemical and geometric features representing these interactions highlighted specificity-conferring features with potential mechanistic insight. Univariable comparative analyses with weighted Wilcoxon-Mann-Whitney tests revealed strong statistical associations between binding site features and specificity that are conserved across unrelated lectin binding sites. Multivariable modeling with random forests demonstrated the utility of these features for predicting the identity of bound glycans based on generalized patterns learned from non-homologous lectins. These analyses revealed *global* determinants of lectin specificity, such as sialic acid glycan recognition in deep, concave binding sites enriched for positively charged residues, in contrast to high mannose glycan recognition in fairly shallow but well-defined pockets enriched for non-polar residues. Focused *fine specificity* analysis of hemagglutinin interactions with human-like and avian-like glycans uncovered features representing both known and novel mutations related to shifts in influenza tropism from avian to human tissues. As the approach presented here relies on co-crystallized lectin-glycan pairs for studying specificity, it is limited in its inferences by the quantity, quality, and diversity of the structural data available. Regardless, the systematic characterization of lectin binding sites presented here provides a novel approach to studying lectin specificity and is a step towards confidently predicting new lectin-glycan interactions.

and featurized representations of the lectin-glycan interactions used in these analyses. The processed data are also provided in the supplemental information. These files were created from publicly available PDB files, listed in the supplementary information, using open-source software and the scripts found in this GitHub repository. The necessary steps to repeat this analysis are outlined in the repository. By request of the original author of code to generate the 3DZDs, we are unable to distribute the original source code or our modifications at this time. However, the compiled binary used to calculate the 3DZDs is available in the main repository and can be used to fully recreate the feature generation process. The modified version of the PLIP tool used is available at https://github.com/demattox/plip.

**Funding:** This work was funded in part by the following grants: U.S. National Library of Medicine T32LM012204 (DEM); Burroughs Wellcome Fund Institutional Program Unifying Population and Laboratory Based Sciences (DEM); National Institute of General Medical Sciences 2R01GM098977 (CBK). The funders had no role in study design, data collection and analysis, decision to publish, or preparation of the manuscript.

**Competing interests:** The authors have declared that no competing interests exist.

**Abbreviations:** 3DZDs, 3-dimensional Zernike descriptors; AAL2, *Agrocybe aegerita* lectin 2; Ca$^{2+}$, Calcium; CFG, Consortium for Functional Glycomics; Fuc, Fucose; Gal, Galactose; GalNAc, N-acetylgalactosamine; Glc, Glucose; GlcNAc, N-acetylglucosamine; Lac, Lactose; LacNAc, N-acetyllactosamine; NeuAc, N-acetylneuraminic acid; NMR, Nuclear magnetic resonance; PDB, Protein Data Bank; PLIP, Protein-Ligand Interaction Profiler; RF, Random forest; RFU, Relative Fluorescence Units; SNFG, Symbol Nomenclature for Glycans; SMILES, simplified molecular-input line-entry system; WMW test, Wilcoxon-Mann-Whitney test (also known as Wilcoxon rank-sum test or Mann-Whitney-U test).

## Author summary

Glycans are sugar molecules found attached to proteins and lipids and coating the outsides of cells from most organisms. Specific recognition of glycans by proteins called lectins facilitates many biological processes, for example enabling influenza virus to gain access to cells, helping the immune system recognize pathogens, and sorting newly built proteins for transport to appropriate cellular regions. Understanding what makes a particular lectin recognize a particular glycan over the vast set of other glycans can help us better understand these processes and how to monitor and control them. To that end, we systematically characterized the sites on lectin structures where glycans are bound, breaking down molecular structures into a comprehensive set of biochemical and geometric features summarizing the sites. This enabled us to discover statistical relationships between binding site features and the glycans recognized by the sites, and further to be able to predict, from a lectin structure, which glycans it recognizes. For the first time, we are able to demonstrate that there are general features of lectin binding sites correlated with and predictive of their specificities, even in unrelated lectins. Ultimately, these findings can help us discover and engineer new lectins for use in research, diagnostics, or even therapeutics.

## 1 Introduction

Lectins, non-enzymatic, non-immunoglobulin, sugar-binding proteins, selectively interact with small subsets of the vast set of possible glycoforms and thereby facilitate diverse biological processes. Minute differences in glycan structure can have profound impacts in associated biological processes. For example, the difference between α2,3-linked and α2,6-linked terminal N-acetylneuraminic acid (NeuAc) glycans serves as the primary barrier blocking avian influenza A from accessing cells in the upper respiratory tract of humans, based on the specificities of the influenza hemagglutinin (HA) [1, 2]. Specific interactions between lectins and their cognate glycans play critical roles in many other host-pathogen interactions [3] as well as an increasing number of known intracellular and extracellular biological processes with altered glycosylation in cancer cells contributing toward tumor cell growth, proliferation, migration, and invasion [4, 5]. Lectins with well-characterized glycan specificities can be leveraged in biomedical applications such as cancer biomarkers [6, 7], cancer therapeutics [8, 9], antiviral therapeutics [10, 11], and drug targeting [12]. Specific lectin-glycan interactions also enable fundamental glycobiology research by tracking and investigating glycans on cells or viruses, in tissues, or in biological samples ranging from blood to human milk, through the use of lectins in mass spectrometry capture strategies, lectin arrays to assess whole cell glycosylation patterns, and labelled lectin probes [13–17].

Applications of lectins are numerous but limited by the specificities of well-characterized lectins. One example of a direct impact of this limitation on glycobiology research is that O-GlcNAcylation, an important but subtle post-translational modification [18], was not discovered until the 1980s [19, 20] and has received disproportionately less research interest compared to other important glycoforms, in part due to the lack of lectin probes efficiently and specifically targeting O-GlcNAcylation until very recently [21, 22]. Novel lectins with novel specificities for known (and currently unknown) glycans will enable even further application of lectins in research, diagnostic, and therapeutic contexts. Sources of novel lectins and lectin specificities include the continued screening of natural products and gene products with potential carbohydrate recognition motifs for sugar-binding activity against target glycans [23] as well as specificity engineering to confer new glycan-binding preferences upon existing lectin

scaffolds [24, 25]. Lectin specificity engineering efforts to date typically rely on extensive study of highly similar lectin binding sites and high-throughput evaluation of engineered variants, but lectin engineering approaches can be advanced with further computational study of glycan specificities of lectin and protein scaffolds [25]. A more thorough understanding of lectin specificity would also facilitate prioritizing putative lectins for characterization of binding activity with glycans of interest, especially if identified specificity determinants can be used to predict lectin specificities. This need is ever-growing with the rapid expansion of genome sequencing capabilities in the past decades and almost 1 million predicted lectins identified from the genomes of over 24,000 species curated within the LectomeXplore database at the time of writing [26].

Experimental investigations into lectin specificities have evolved considerably from initial efforts to characterize lectins based on their abilities to agglutinate blood cells and subsequent competitive inhibition approaches with defined glycans [27]. With increased control over glycan synthesis mechanisms and production, diverse and well-defined glycoforms are increasingly available for use in characterizing lectin-glycan interactions in more detailed experimental approaches, including isothermal titration calorimetry and equilibrium dialysis [27], higher-throughput approaches including surface plasmon resonance [28] and frontal affinity chromotography [29], and the highest-throughout approach of glycan microarrays which simultaneously characterize a large number of lectin-glycan interactions [27, 30–32]. However, while these methods provide clear pictures of *which* features of bound glycans a given lectin will specifically recognize, they do not address the question of *how* lectins specifically recognize some glycans but not others. To this end, structural characterizations through X-ray crystallography and nuclear magnetic resonance (NMR) are able to identify glycan-binding sites of lectins as well as the residues and structural features conferring specific interactions. These methods unfortunately still suffer from serious limitations; in addition to time and expense, they generally are not able to accurately resolve larger glycan structures [33].

Information-rich experimentally-determined structures have been further computationally analyzed and leveraged in wide-ranging studies of lectin-glycan interactions. Molecular dynamics (MD) simulations have been employed with great success to gain detailed, mechanistic understandings of individual lectin-glycan interactions [34, 35], but MD approaches are limited by requirements of time, expertise, and computational resources which prevent broader utilization and higher-throughput probing of potential interactions between lectins and glycans. There have been several other efforts to systematically compare and analyze protein-glycan interactions in the Protein Data Bank (PDB) [36–38]; however none to date have focused on specificity or uncovered interpretable features contributing to glycan-binding preferences. Briefly, GlyVicinity [36] calculates the frequencies of amino acids within a set distance of a carbohydrate residue from all available PDB entries and compares these frequencies to background amino acid frequencies in order to highlight enriched or depleted residues, but it does not take into account the context of the lectin binding site containing the amino acids or the glycan containing the monosaccharide residue. Shanmugam et al. [37] predict protein-carbohydrate binding affinity using features derived from the binding site, the glycan, and the interactions between the molecules, but utilizes a fairly limited set of interactions and does not investigate potential determinants of specificity within the selected features. Finally, Cao et al. [38] perform pairwise comparisons for protein-carbohydrate interactions within the PDB, measuring the structural similarity of the binding sites and the similarity of the interaction patterns, but without systematically identifying binding site features or relating characteristics to observed specificity or promiscuity.

In this study, we seek to identify specificity-determining features of lectin binding sites through systematic characterization and comparison of lectin structures solved in complex

with glycans that they recognize. By leveraging curated lectin-glycan complex structures compiled in the UniLectin3D database [39], we are able to detail significant and predictive features of lectin binding sites associated with their ability to accommodate given glycans compared to binding sites interacting with other glycans. This assessment of lectin specificity is considered for the most commonly occurring glycans, along with three naturally defined groups of glycans (terminal NeuAc glycans, high mannose glycans, and terminal fucose glycans) with high biological relevance, especially in human health [40–45]. From structural analysis, statistical characterization, and predictive modeling of over 4,000 lectin-glycan interactions, particular sets of features are found to be significantly associated with global lectin specificity compared to background interactions for these glycans, with many of the significant features also having high importance in predictive classifiers capable of identifying a bound glycan from its interaction site features. In general, these features reveal conserved and distinguishing patterns in lectin binding sites with overlapping specificities, supporting specific observations such as the basis for similarity in lectin recognition of N-acetylglucosamine and galactose compared to that of non-acetylated glucose. A further investigation of fine specificity of influenza hemagglutinin structures with human and avian glycans highlights both known and novel mutations contributing to recognition. These findings demonstrate the utility of this systematic, structural approach to study lectin binding site structures in providing a strong basis for the longer-term goal of predicting novel lectin-glycan interactions and rationally engineering lectin specificity.

## 2 Results

In order to discover and evaluate the utility of molecular determinants of specific lectin-glycan recognition (Fig 1), we comprehensively "featurize" a large set of experimentally determined glycan-bound lectin structures, with each individual occupied binding site further referred to as an "interaction". We subject these interactions to complementary univariable comparative analysis and multivariable predictive modeling in order to investigate *global* lectin recognition of certain glycans compared to all other glycans. In the following, we begin by summarizing the lectins, glycans, and interaction features supporting the study (subsection 2.1, Fig 1B–1D). We then examine the contributions of these features to global lectin-glycan recognition specificity (Fig 1E). To this end, we first characterize general findings from univariable statistical analysis, demonstrating significant differences in interaction sites containing each glycan individually when compared to lectin interactions with all other glycans (subsection 2.2). We then complement this analysis with multivariable predictive modeling for each glycan, showing that combinations of features are often able to reliably predict whether the lectins are recognizing the glycan of interest or another glycan (subsection 2.3). We next elaborate the different groups of features discovered in the univariable and multivariable analyses and elucidate global determinants of specificity for one glycan vs. others (subsection 2.4). Separately, we demonstrate the utility of these interaction characterizations for study of nuanced differences in specificity by investigating the determinants of *fine* hemagglutinin specificity, comparing α2,6-linked terminal NeuAc interactions directly to α2,3-linked glycan interactions and uncovering associations with greater sensitivity than could be achieved with comparison to background interactions (subsection 2.5, Fig 1F).

### 2.1 Data collection and interaction characterization

Our analysis of lectin-glycan specificity was based on a large set of co-crystal structures curated for quality and ligand-identity, relying on co-occurrence in solved structures as an assessment of specificity. A list of PDB IDs for lectin-glycan structures, along with associated information about the lectin and glycan, was obtained from the UniLectin3D database [39]. Non-glycan

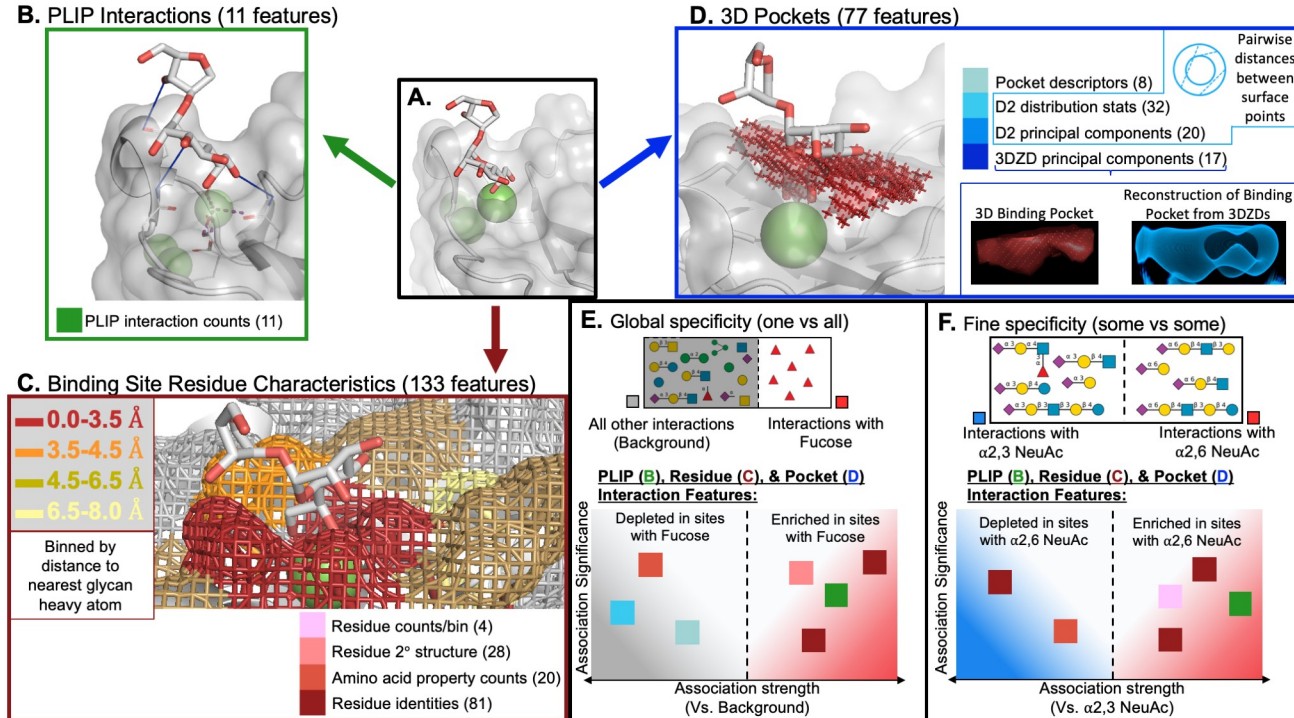

**Fig 1. Lectin-glycan interaction characterization and comparison.** Features for lectin-glycan interactions (A) are derived from Protein-Ligand Interaction Profiler (PLIP) defined interaction counts (B), voxelized representations of the 3D pocket space occupied by the glycan (C), and binding site residues binned by their minimum distance to the glycan (D). Two types of specificity analyses were conducted. For *global* specificity (E), binding interaction characteristics from each glycan of interest were compared to the background characteristics of all other lectin-glycan interactions, revealing features that were enriched or depleted in association with the presence of the given glycan relative to all other glycans. For *fine* specificity (F), characteristics were compared among interactions within a subgroup of similar glycans. In panels A-D, the binding interaction between human lung collectin surfactant protein D and a disaccharide fragment (Hep-Kdo) of a bacterial lipopolysaccharide is used to demonstrate the three categories of interaction features (PDB ID: 4E52). Panel C has additional components illustrating featurization of the voxel point cloud via features describing the D2 distribution of pairwise distances between surface points and computed 3D Zernike descriptors (3DZDs), with the original point cloud in red and the reconstructed shape from the 3DZDs in blue. Panels E & F display schematic results of select features defined in panels B-D that were found to be significantly enriched or depleted in the specified interactions. Structures were rendered using PyMol and glycan symbols follow the Symbol Nomenclature for Glycans (SNFG) system.

ligands were eliminated, missing covalent bonds in glycan structures were added, and all suspected glycosylation occurrences were excluded, leaving a curated set of 4,088 lectin-glycan interactions from 1,364 structures representing 412 unique lectins in complex with 226 unique glycan ligands (available in S1 File).

To reduce bias in analysis resulting from redundancy and close homology among lectins, non-redundant protein chains were extracted from each structure (S1 Fig) and the 1,364 structures were clustered at 50% sequence identity. The vast majority of the resulting 225 clusters of homologous/redundant lectin structures contained 5 or fewer unique lectins (by UniProt ID), although the largest clusters had more than 15 unique lectins (S2 Fig), confirming the necessity of this approach to prevent more well-studied lectins from overly influencing studies of specificity. At each step of the analysis, interaction weighting or sampling based on these clusters was applied to prevent disproportionate impact from better-represented lectins in larger homology clusters.

To ensure a sufficient number of diverse interaction examples, only the the 12 most common unique glycan ligands bound to lectins from different homology-based clusters are further considered, as well as three classes of glycans likely to be specifically recognized by lectins:

terminal NeuAc glycans, high mannose glycans, and terminal fucose glycans S3 and S4 Figs. The provided IUPAC identifiers of the individual glycans comprising each class can be found in S1–S3 Tables. Henceforth, the 12 individual glycans and three glycan classes are referred to as the 15 "glycans of interest" and displayed in future figures in the order shown in S4 Fig, arranged by glycan class and prevalence in complex with different lectins.

Interactions between the lectins and glycans of interest (Fig 1A) were represented by a comprehensive set of 221 complementary geometric and physicochemical features separated into three general categories:

- *11 interaction features* (Fig 1B), generated by the Protein-Ligand Interaction Profiler (PLIP) tool [46]), describing the numbers and types of non-covalent interactions including hydrophobic interactions, hydrogen bonds, water bridges, electrostatic interactions, and metal coordination.

- *133 binding site residue features* (Fig 1C) describing the amino acids and associated secondary structures in four separate binned distances from the glycan, as well as the number of calcium ($Ca^{2+}$) ions in the pocket. To account for the flexibility of protein structures and the highly flexible nature of glycans, binned residue representations approximate the probability of interacting with the glycan in other possible low-energy conformations instead of relying on the exact conformation in the solved crystal structure.

- *77 3D pocket geometry features* (Fig 1D) describing the three dimensional space of the interaction pocket where the glycan is found. These features were derived from voxelized representations of the interaction site as characterized by rotationally-invariant 3D Zernike Descriptors (3DZDs) [47, 48] and D2 distributions [49]. Voxelized representations were generated with varied thresholds to better capture the diversity of pocket shapes and sizes, with 3DZD and D2 approaches utilized to represent complex shapes in compact, robust, featurized forms allowing for easy comparison of pocket shapes and sizes. The D2 distributions summarize all pairwise distances between points on the surface of the voxelized pocket representation and features were generated describing the statistics of the distributions as well as principal components capturing more nuanced sources of variation. The 3DZDs compactly describe the shape of the pocket with 3D Zernike moments at a level of detail that allows for the reconstruction (blue) of the original pocket shape (red, Fig 1D).

Taken together, these 221 features enable systematic comparisons between lectin interactions with glycans of interest, identification of determinants of *global* specificity associated with interactions containing a given glycan compared to all other interactions(Fig 1E), as well as determinants of *fine* specificity found in a separate comparison of a interactions to other interactions with similar glycans (Fig 1). Features for each interaction from each lectin structure are available in S1 File.

## 2.2 Lectin binding site features are significantly associated with specific glycans

Given a particular glycan of interest, the patterns in lectin-glycan interactions detected across diverse lectin binding sites indicate determinants of global lectin specificity for that glycan. Comparing lectin interactions containing one glycan with all other lectin-glycan interactions then highlights those features that are enriched or depleted for that glycan. To control for bias from redundant and homologous lectin structures, these comparisons were performed here with weighted Wilcoxon-Mann-Whitney (WMW) tests [50], weighted by the sizes of the groups of homologous lectins as well as the numbers of individual lectins in those groups.

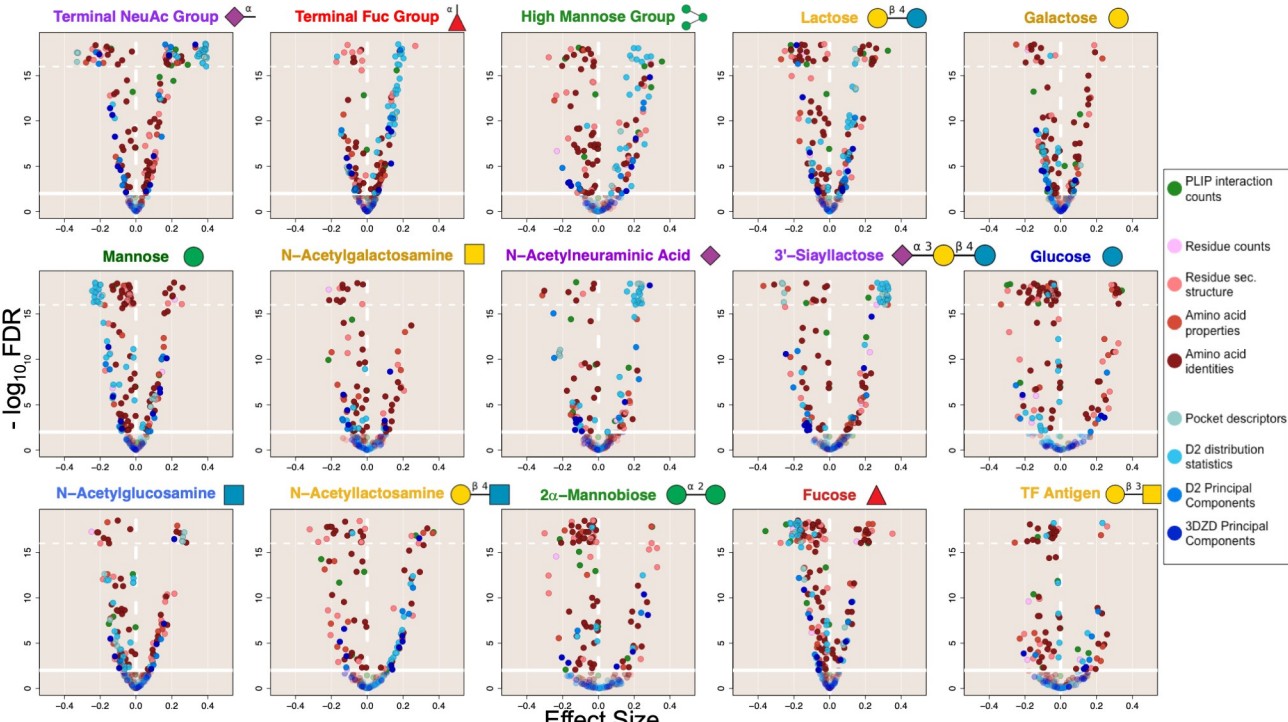

**Fig 2. Lectin binding site features have significant associations with the presence of specific glycans.** Volcano plots show that a substantial proportion of features from all three categories are statistically significantly (q < 0.01) enriched (*x* > 0) and depleted (*x* < 0) in interaction characterizations for each of the 15 glycans of interest when compared to background interaction characterizations from all other glycans. It is apparent that pocket-size-correlated D2 distribution & pocket descriptor features (represented by the two lightest blue colored points) are generally enriched for larger glycan ligands (terminal NeuAc, high mannose, 3'-siayllactose) and depleted for interactions with smaller ligands (monosaccharide glycans). Some glycan-lectin interactions have fewer features that are strongly enriched (terminal fucose, N-acetyllactosamine, and TF antigen), possibly indicating a diversity of interaction mechanisms, or that more common, highly similar glycans in the background are reducing the strength of associations. Significance and direction of association was determined by weighted Wilcoxon-Mann-Whitney (WMW) tests accounting for homologous and redundant lectin structures. The x-axis shows the direction and strength of rank-based enrichment for each feature compared to background. The y-axis indicates the statistical significance (q-values) adjusted by the Benjamini-Hochberg procedure applied separately for each ligand with a significance threshold set for an FDR of 0.01 (represented by the solid horizontal lines). Q-values more significant than $1 \times 10^{-16}$ (horizontal dotted line) were scattered between $3 \times 10^{-19}$ and $1 \times 10^{-16}$. The vertical line (*x* = 0) divides positive (right) and negative (left) associations. Glycan symbols follow the SNFG system.

The results from the 15 WMW tests conducted for the glycans of interest (Fig 2) show that there were large, diverse sets of interaction features significantly associated with each of these glycans when compared to interactions involving all other glycans. The particular features are discussed in detail in subsection 2.4, but some trends are already obvious. For example, points colored with the two lightest shades of blue, corresponding to features describing the pocket voxelization and the statistics of its pairwise-distance D2 distribution, appeared as a group and were generally enriched or depleted together. Since these features were both influenced by the overall size and volume of the interaction pocket, and since the interaction site was defined in part by the size of the glycan, it is unsurprising that these features were strongly depleted in interactions with monosaccharides such as mannose, glucose, and fucose, while being strongly enriched in interactions with larger glycans such as the terminal NeuAc group, terminal fucose group, high mannose group, lactose, and 3'-siayllactose (NeuAc(a2–3)Gal(b1–4)Glc). Interestingly, this enrichment was very strong for N-acetylneuraminic acid despite its being a monosaccharide.

Some glycans, including the terminal NeuAc group, the high mannose group, Lac (lactose, Gal(b1–4)Glc)), mannose, glucose, and fucose, manifested many significant associations with

large effect sizes, both positive and negative, with varied lectin binding site features. It can be interpreted that there are conserved geometric and physicochemical features in the lectin binding sites that specifically recognize these glycans, thereby representing determinants of global specificity. On the other hand, some glycans, most notably TF antigen (Gal(b1–3)GalNAc), had fewer interaction features enriched above background; similarly 2α-mannobiose (Man (a1–2)Man), N-acetylglucosamine, and N-acetylgalactosamine also appear to have had fewer significantly enriched features. This trend is potentially attributable to a diversity of binding mechanisms in the observed interactions leading to reduced significance, especially in case of the terminal fucose group which included a large diversity of glycoforms (S3 Table) and fewer significantly associated features than fucose monosaccharide. Another explanation for reduced feature enrichment in the other cases is the presence of other similar glycans in the background, reducing the significance and degree of enrichment of shared interaction features and compounded in cases where potentially similar glycans (such as Lac and LacNAc (N-acetyllactosamine, Gal(b1–4)GlcNAc)) are much more prevalent and are recognized in similar interactions, adding similar examples to the background and reducing association strengths for the less prevalent glycans (such as TF antigen) (S4 Fig).

## 2.3 Lectin-glycan interaction features are predictive of the identity of the bound glycan

While univariable comparative analysis revealed that there were indeed specific lectin binding pocket features associated with specific glycan recognition, it did not (and cannot) characterize the extent to which combinations of these features generalize to new cases and are thereby actually *predictive* of which glycans a particular lectin will recognize. Thus multivariable predictive modeling, in particular supervised classification, complements the univariable comparative analysis by demonstrating that in some cases, particular feature combinations suffice to predict specific recognition. Here, the classification goal was to train, for each glycan of interest, a glycan-specific model that labels each lectin structure as "positive" (the glycan is actually bound in the structure) or "negative" (a different glycan is bound) based on combinations of binding site features learned from training data involving other, distinct interactions. Random forest (RF) classification models [51] were used because of their interpretability as well as suitability for high-dimensional data without detrimental impact from collinearity. RF models for each glycan of interest were validated with a leave-one-out approach: binding-site structures from one of the homologous lectin clusters were withheld, a model was trained on sampled dissimilar binding-site structures from the remaining lectins, and then classification performance was evaluated on selected, dissimilar examples from the withheld structures. We note that while a "negative" label could mean that the glycan and lectin do not interact, it could also mean that, while the pair actually does interact, that interaction is solved in a different structure or the structure has yet to be solved. For this reason, the prediction performance was evaluated separately for both *recall*, the fraction of the lectin structures with that glycan bound that are correctly predicted to be positive, and *precision*, the fraction of the lectin structures predicted to include that glycan that actually do; note that these performance metrics are not impacted by the true negative rate. Additionally, models were trained using $F_2$ scores to combine precision and recall with a greater weight on recall since recall only accounts for positively labelled data.

The distributions of recall and precision for each glycan from repeated leave-one-out cross-validation are represented by violin plots in Fig 3 and compared to the performance of corresponding "null model" RFs trained and validated in the same manner but using interactions with shuffled glycan labels and thus expected to display essentially random performance

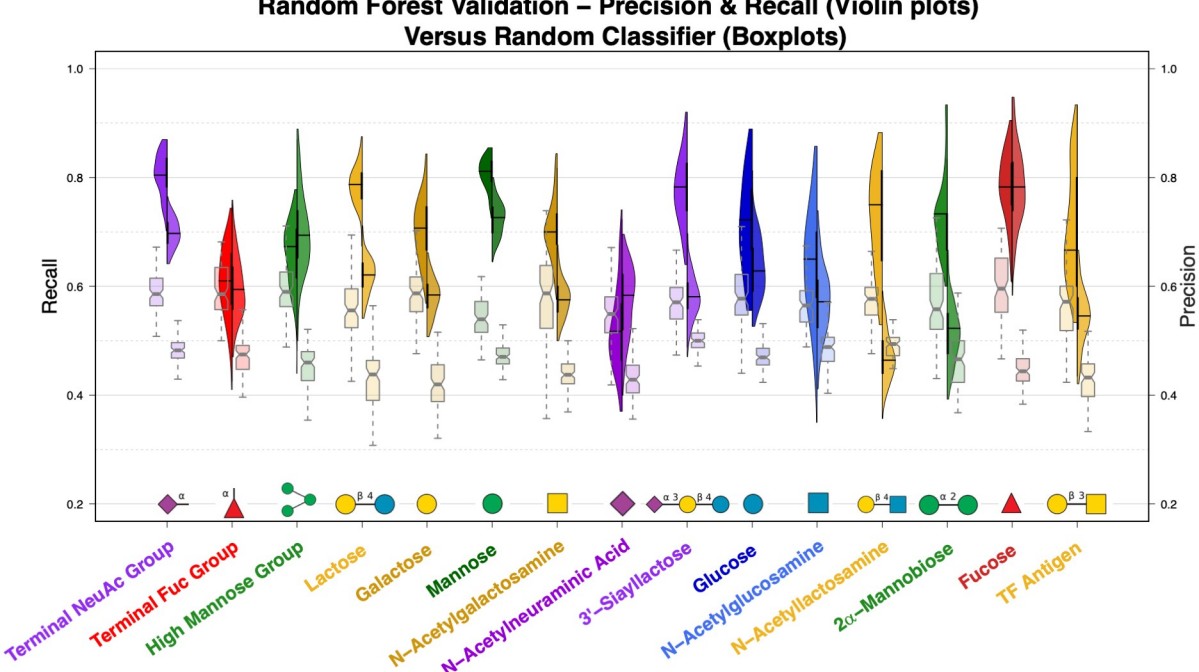

**Fig 3. Lectin binding site features can be used to predict the identity of bound glycans.** Random forest models trained for each of the 15 glycans have strong recall performance while predicting whether interactions contain the respective glycan based on the interaction features alone. The models are predictive of glycan identity even when trained only on lectins with less than 50% sequence identity, outperforming identical classifiers trained on data with shuffled labels. Split violin plots show the recall (left-hand distribution and left y-axis) and precision (right-hand distribution and right y-axis) of ligand-specific random forest models measured during leave-one-out cross-validation. The pairs of notched boxplots for each glycan show the performance of classifiers trained on data with shuffled labels, where again the left-hand boxplots depict recall and the right-hand boxplots depict precision. Glycan symbols follow the SNFG system.

(shown as boxplots). The glycans' prevalences in complex with different lectins were generally proportional to the amount of training data (S4 Fig). Overall, the classifiers performed well for both recall and precision (mean values of 0.71 and 0.61 respectively) as compared to recall and precision of the classifiers trained with shuffled labels (mean values of 0.57 and 0.46 respectively). Performance on the training data was very similar to the cross-validated performance (S5 Fig, mean recall = 0.70 & mean precision = 0.64), indicating that overfitting was not likely, and performance as measured by the $F_2$ score positively correlated with the number of interactions available for training ($\rho = 0.39$, p < 0.001), with fucose-predicting models doing especially well despite having had relatively few training samples Fig 3 and S6 Fig. Broader distributions of performance metrics in Fig 3 indicate that model performance was more sensitive to the sets of dissimilar interaction examples randomly sampled for use in training and validation.

The RF models did very well for NeuAc terminal glycans, mannose monosaccharide, and fucose monosaccharide, with all median recall values above 0.78 and median precision values above 0.69. For these glycans, the associated lectin binding site features can be used to easily detect interactions, verifying the value of these features in studies of specificity with some of the most predictive features shared by these three models including the relative abundance of charged polar amino acids in the residues closest to the glycans as well as 3D pocket features correlated with the size of the interaction site. In light of the discussion above regarding positive/negative classification labels, the high precision of these models can be interpreted to mean that the lectins binding these glycans are not often crystallized in complex with other

similar glycans, and most of the lectins capable of binding these glycans are crystallized with these glycans.

In many cases, including lactose, galactose, N-acetylgalactosamine, 3'-sialyllactose, glucose, LacNAc, and 2α-mannobiose, the glycan-specific models maintained high recall despite having lower precision, with median recall values of at least 0.70 while median precision values ranged from 0.46 to 0.63 in this group. Strong recall performance indicates these predictive models are still able to recognize the glycans of interest from the physicochemical and geometric characterization of the interaction site (features that appear to be particularly predictive are discussed in the next section). However, the lower precision of these models can be attributed to the same lectins appearing in complex with other glycans, particularly with similar glycans. For glycans recognized by lectins that interact with other similar glycans, this effect was more pronounced among the less prevalent glycans than among their more common counterparts, e.g., LacNAc had a median recall of 0.75 but also the lowest median precision (0.46), in contrast to lactose, which had a slightly higher median recall (0.79) but a much better median precision (0.62) (Fig 3 and S4 Fig).

Interestingly, the RF classifier for the high mannose glycan group had higher precision than recall, with both outperforming the null model. While recall for the high mannose classifiers was slightly better than that for the models trained on shuffled labels, median precision was the 4th highest. This might indicate that high mannose glycans are recognized by a number of diverse binding mechanisms with some shared underlying commonalities that make the model precise enough to eliminate other interactions but not strong enough to reliably recognize all of the high mannose interactions. This observation is not as immediately apparent from the statistical comparisons in the previous section, demonstrating the strength of complementing the comparative analyses with predictive modeling.

For the remaining glycans, N-acetylneuraminic acid, N-acetylglucosamine, the terminal fucose group, and TF antigen, recall and precision were only marginally better, if at all, when compared to the "null model" classifiers trained on shuffled labels. In these cases, it is likely that a diversity of interaction mechanisms are present, especially in the case of terminal fucose glycans as mentioned previously, and the RF models were not able to learn conserved patterns in the interaction features sufficient to reliably recognize binding of these glycans. This finding demonstrates the limitation of using co-occurrence in crystal structures as a model of specificity, a point elaborated in section 3.

## 2.4 Significant and predictive features reveal global determinants of specificity

Lectin interaction features that were both significant and highly predictive across diverse lectins and conserved across interactions with similar glycans are likely to play a role in facilitating lectin specificity. By integrating the discovered features from the comparative and predictive approaches, we thus aim to obtain higher confidence in the identified features and a better basis for deriving possible explanations for trends seen in the analyses. Fig 4 illustrates similarities and differences among glycans in terms of enrichment and depletion of interaction features, with particular observed relationships (boxed) discussed in more detail in the remainder of this section. In summary, the figure shows the results using all 221 features (Fig 4A) as well as the different types of features (Fig 4B–4D), using color to show the WMW effect size values of the features and bullet points to call out the features that were significant in the univariable analysis (q < 0.01) and highly predictive in the multivariable analysis (75th percentile by median ranked feature importance for their respective feature type). Glycans were clustered according to their WMW profiles, so that in panel (A) they are grouped together when lectins

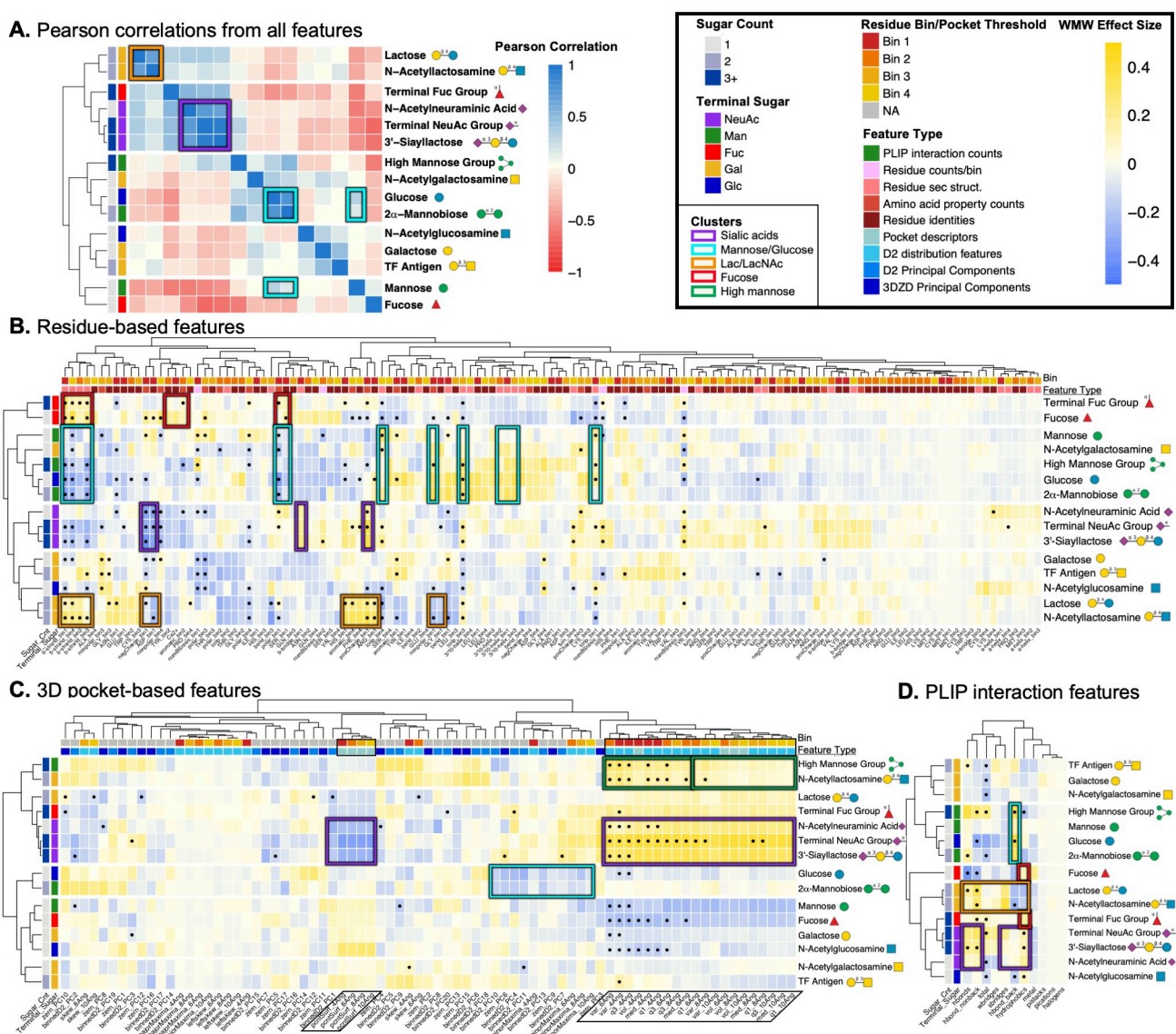

**Fig 4. Determinants of global lectin specificity are shared for similar glycans.** Similar glycans have similar patterns of enriched and depleted interaction features as observed by Pearson correlations between weighted WMW feature effect sizes. Panel A shows the correlogram from all 221 interaction feature effect sizes, clustered by Pearson correlation coefficient. Panels B-D show heatmaps of the interaction feature effect sizes with features in the columns and ligands in the rows clustered by Pearson correlation. Features that are statistically significant by the weighted WMW tests (q < 0.01) and in at least the 75th percentile of median feature-type-stratified importance from the random forest models are indicated with bullet points. The color bars present along the columns indicated the subcategory of the feature and the parameter threshold used when extracting the feature. The color bars along the rows indicate the identity of the terminal saccharide in the glycan and the number of saccharides present. Clusters discussed include sialic acid glycans (purple boxes), mannose and glucose (cyan boxes), lactose and N-acetyllactosamine (orange boxes), and fucose and terminal fucose containing glycans (red boxes). Interestingly, N-acetylglucosamine interactions are more similar to interactions with galactose while N-acetylgalactosamine interactions are more similar to interactions with glucose. The dark green boxes indicate distinct patterns in the 3D pockets of interactions with high mannose. Glycan symbols follow the SNFG system.

recognize them similarly in terms of overall pocket features, while in panel (B) the relationships reveal shared physicochemical environments and recognition motifs, in panel (C) the clustering highlights 3D geometry-based relationships, and in panel (D) clustering is based on similarities in PLIP-characterized atomic interactions. The definition of "highly predictive" features was made so as to mitigate enriched collinear features from one group outweighing

predictive features from others (S7–S10 Figs). These stratified ranked feature importances positively correlated with the absolute value of the WMW effect size ($\rho = 0.35$, $p < 0.01$), with stronger correlation observed especially for glycans with better-performing random forest models (S11 Fig).

**Determinants of specificity are shared between lectin interactions with similar glycans.** There are three clusters of lectin-glycan interactions that were similar for the different subsets of the features and are thus highlighted with colored boxes in Fig 4: a sialic acid cluster (purple boxes), a mannose/glucose cluster (cyan boxes), and a lactose/LacNAc cluster (orange boxes). Groupings of glycans by lectin interactions deviated slightly in Fig 4C due to the strong influence of the size of the glycan ligand on the extent and characterization of the interaction site; in this case the primary factor that appears to drive clustering is ligand size and therefore the overall size of the 3D interaction space.

**Sialic acid cluster** (Fig 4, purple boxes) Interactions with sialic acid glycans, i.e., NeuAc monosaccharide, 3'-sialyllactose, and the terminal NeuAc group, were the most tightly correlated cluster and the tight grouping is strongly conserved across feature types as well. The highest observed pairwise correlation for any glycans came from 3'-sialyllactose and the terminal sialic acid group ($\rho = 0.91$, $p < 0.001$). While this is not particularly surprising as 3'-sialyllactose is one of the 27 terminal NeuAc glycans, NeuAc monosaccharide is not but still has strong correlations with both of the other sialic acid glycans ($\rho = 0.71$, $p < 0.001$ for terminal NeuAc glycans & $\rho = 0.68$, $p < 0.001$ for 3'-sialyllactose) (Fig 4A). In summary, these interactions shared a strong enrichment of positively charged residues and depletion of negatively charged residues near the glycan (unsurprising for negatively charged glycans), a general enrichment of β-bridges over other secondary structures especially β-strands, a very large 3D interaction space around the glycan, and a strong enrichment of the number of hydrogen bonds (especially side chain mediated) as well as hydrophobic interactions, with a slight enrichment of electrostatic interactions. These association patterns indicate these sialic acid glycans are typically bound by large lectin binding sites capable of making many hydrogen bond and hydrophobic interactions where positively charged residues seem to be present due to charge complimentary without participating in substantially more electrostatic interactions compared to background lectin-glycan interactions.

**Mannose/glucose cluster** (Fig 4, cyan boxes) The high recall and low precision of the RF classifier for 2α-mannobiose indicated other prevalent glycans are likely recognized by lectins in a similar fashion, which was confirmed by a very strong correlation between the interaction features from 2α-mannobiose and glucose ($\rho = 0.77$, $p < 0.001$), weaker correlation with mannose monosaccharide ($\rho = 0.31$, $p < 0.001$), and a general trend of glucose clustering with at least one of the three mannose glycans for each feature type, somewhat intuitively since mannose is a C-2 epimer of glucose. In summary, this mannose-glucose clustering appeared to be driven by very similar interaction pockets for 2α-mannobiose and glucose, depletion of β-strands and polar residues in favor of enriched non-polar residues, $3_{10}$ helices, as well as loop structure, and a general depletion of all other interaction types except for backbone hydrogen bonds. Taken together, this paints a picture of mannose recognition requiring specific secondary structure arrangement to coordinate backbone hydrogen bonding with primarily non-polar amino acids.

**Lactose/N-acetyllactoseamine cluster** (Fig 4, orange boxes) Similar lectin recognition of Lac and LacNAc was proposed as an explanation for the high recall but low precision of their RF classifiers, somewhat intuitively as they differ by a single acetyl group on the reducing terminal sugar. Feature effect sizes for these two were indeed among the most strongly

correlated ($\rho = 0.70$, p < 0.001), driven by a depletion of hydrophobic interactions, backbone hydrogen bonds, and negatively charged residues in favor of side chain hydrogen bonds and electrostatic interactions facilitated by an enrichment of positively charged residues, asparagine in the closest bin to the glycan, and β-strand secondary structure. Lactose and LacNAc specific lectin binding sites appear to utilize positively charged residues and select polar residues to coordinate hydrogen bonds via side chain groups and electrostatic interactions with the charge center at the glycosidic bond of these disaccharides. Additionally, Lac and LacNAc motifs often present terminal sialic acids and are recognized together as larger binding epitopes by many lectins. This association is seen in the overall positive correlation between features associated with Lac/LacNAc interactions and with interactions included in the sialic acid cluster.

**Despite diverse interaction mechanisms, terminal fucose is still recognized similarly to fucose monosaccharide.** While the terminal fucose group's set of glycoforms was likely too diverse to permit more significant associations according to our criteria, the features that were conserved and important for recognizing these diverse glycans were generally shared with fucose monosaccharide, confirming the importance of these features for specific recognition of fucose and fucose-terminal glycans compared to other glycans. These shared features included a depletion of non-polar residues, aromatic residues (especially tyrosine) in the bin closest to the glycan, and generally all secondary structure besides β-strands; along with an enrichment of hydrophobic interactions, β-strands, polar residues (especially serine) in the closest bin to the glycan, and aromatic residues (including tyrosine) in the next closest bin (Fig 4B–4D, highlighted by red boxes). In summary, these findings portray lectin recognition of fucose relying on polar residues in close proximity to the glycan and tyrosine/aromatic residues slightly further away, often found within β-strands, to coordinate numerous hydrophobic contacts with the glycan.

**Size-correlated collinear features still differentiate between large lectin pockets with different specificities.** As highlighted above, the 3 sialic acid glycans of interest in this study were recognized by lectins in similar ways. However, it is remarkable that sialic acid monosaccharides had 3D pocket features that were very similar to those of the other much larger sialic acid glycan ligands (Fig 4D) since the extent of the 3D pocket is heavily dependent on the size of the glycan ligand and 3D interaction pockets of fucose and mannose monosaccharides were very distinct from their respective groups of larger glycans (terminal fucose and high mannose glycans). This indicates that the binding sites from sialic-acid-recognizing lectins were generally well defined by the presence of a single NeuAc monosaccharide and the size & extent of the interaction site is robust to the size of the glycan ligand. Representative structures further illustrate this interpretation, demonstrated by interactions between a terminal NeuAc glycan and murine polyomavirus (Fig 5A) and between NeuAc monosaccharide and influenza hemagglutinin (Fig 5B). The depicted representative interactions were selected as being the closest to the weighted feature-specific means for the glycans of interest. For both interactions, the binding sites were long, wide, and fairly concave, such that the NeuAc monosaccharide was able to fit fully in the interaction pocket and the voxelized representations captured the pocket to the same extent as if a larger glycan were present.

While lectins interacting with high mannose glycans also had much larger interaction sites than most of the glycans of interest, these binding sites were much more compact and shallow compared to sialic-acid-specific binding sites, as can be seen in the representative interaction between a high mannose glycan and concanavalin A (Fig 5C). This observation was also apparent in the pocket-size features in the second cluster from the top in Fig 4C highlighted in a dark green box. These pocket-size features from the smaller thresholds (left-most dark green

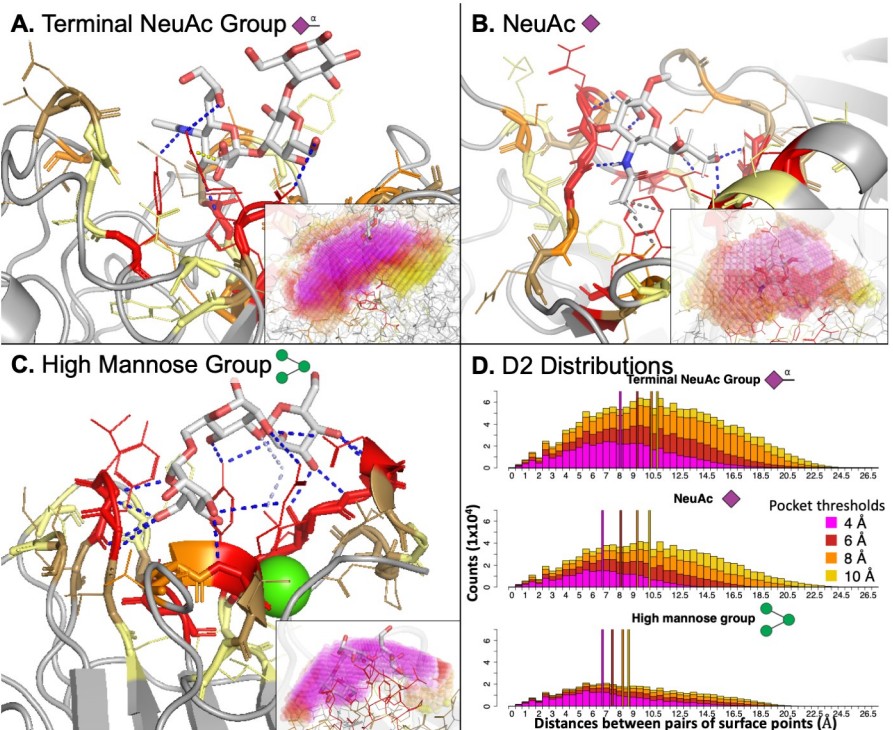

**Fig 5. Sialic acid recognizing lectin binding sites are much deeper and more concave than the fairly flat and shallow binding sites of lectins that bind high mannose.** Representative lectin interactions with a terminal NeuAc glycan (panel A, PDB ID: 1SID), NeuAc monosaccharide (panel B, PDB ID: 1HGH), and high mannose (panel C, PDB ID: 1CVN) demonstrate the differences in the 3D interaction site space between NeuAc-binding lectins and high-mannose-binding lectins. Panel D shows the D2 distributions summarizing pocket geometry for each of these representative interactions. The lectin binding sites containing sialic acid glycans are wider and more concave while the high-mannose-accepting binding sites are more shallow and compact, being nearly entirely defined by the lowest threshold used for pocket generation as seen in the inset subpanels in A-C and in the D2 distributions in panel D. In panels A-C, residues are colored by their binned distance from the glycan (red: bin 1, orange: bin 2, sand: bin 3, pale yellow: bin 4), the glycan is colored by atom-type with carbons in white, and the rest of the lectin structure is in grey. PLIP interactions are colored blue for hydrogen bonds, pale blue for water bridges, yellow for electrostatic interactions, and grey for hydrophobic interactions. In the insets, 0.5 Å$^3$ spheres were placed at each voxel center in the pocket and colored by the distance threshold used (magenta/red/orange/yellow: 4/6/8/10 Å). In panel D, vertical lines were placed at the median D2 measure from each threshold with the same coloring as used from the insets in panels A-C. All structures were rendered in PyMol and glycan symbols follow the SNFG symbols.

box) used to voxelize the pocket (4 & 6 Å) were more strongly enriched and predictive compared to the same features from pocket representations built with larger thresholds (8 & 10 Å, right-most dark green box). Thus while the interaction space around these large glycans was larger than seen in the background interactions, it was fully defined by the voxelization with the two smallest thresholds, and considering the portion of the pocket further from the glycan did not substantially aid in characterizing the interaction. This was confirmed in the D2 distributions for these three representative interactions in Fig 5D, where the number and length of the pairwise distances continued to grow for the sialic acid glycans, while the high mannose interaction was almost entirely defined by the representation with the lowest threshold (4 Å, magenta).

**Lectins differentially recognize Glc compared to GlcNAc & Gal compared to GalNAc.** It appears that interactions with GlcNAc were more strongly correlated with lectin recognition of galactose and other non-reducing terminal galactose glycans, while interactions with GalNAc were more often clustered together with glucose and mannose glycans (Fig 4A–4C).

Interaction feature associations from the acetylated derivatives were weakly positively correlated with their opposing non-acetylated counterparts ($\rho$ = 0.37, p < 0.001 for Glc/GalNAc, $\rho$ = 0.32, p < 0.001 for Gal/GlcNAc), but the association was much stronger compared to the correlations with their corresponding non-acetylated counterparts ($\rho$ = 0.15, p < 0.001 for Gal/GalNAc, $\rho$ = -0.14, p < 0.05 for Glc/GlcNAc).

This finding is informative in the search for novel and improved probes for O-linked N-acetylglucosamine (O-GlcNAcylation) modifications, for which there were not any appropriate lectin probes until the very recent discovery and characterization of the terminal-GlcNAc specific fungal lectin *Agrocybe aegerita* lectin 2 (AAL2) [22, 52]. The observed trends in determinants of global specificity for GlcNAc indicate that additional novel probes for the study of O-GlcNAcylation might be more easily found or engineered from galactose-binding families of lectins than from glucose-binding lectins. In fact, Consortium for Functional Glycomics (CFG) glycan microarray results for AAL2 at the highest concentration used by Jiang et al. [52] showed strong specificity for non-reducing terminal GlcNAc, but 5 of the top 50 (> 90th percentile rank) bound glycans had a non-reducing terminal galactose residue (ranked orders: 32, 42, 44, 48, & 50) while none of the 8 available non-reducing terminal glucose glycans appeared have greater than 10 Relative Fluorescence Units (RFU) (< 50th percentile rank), further confirming this association. Fine specificity comparisons between interactions with terminal GlcNAc and interactions with terminal galactose could likely further elucidate determinants of specificity for GlcNAc over galactose and provide initial direction for engineering GlcNAc-specificity in galactose-binding lectins.

## 2.5 Known and novel determinants of fine influenza hemagglutinin specificity

Influenza hemagglutinin (HA) is a very well-studied lectin due to its critical role in mediating influenza infections by targeting 6' αNeuAc-terminal glycans in the upper respiratory tracts of humans for viral entry. The fine specificity of influenza HA proteins, in particular distinguishing the recognition of human-like α2,6- versus avian-like α2,3-NeuAc-terminal glycans, is also well studied. It has been shown through detailed, manual comparisons of crystal structures and HA sequences that a few mutations in HA binding sites can shift specificity and enable pandemic influenza strains to jump from avian populations and wreak havoc in immunologically naive human populations [1, 53]. We here complement those studies by applying our systematic analyses and comparisons in a deeper investigation to characterize fine HA specificity distinguishing these two similar but critically different glycans. The univariable analysis approach is similar to that for global specificity, but now comparing these glycans' interactions directly with each other, rather than against the background of all others. Due to the limited amount of data available for each set, the previously employed predictive modeling approach could not be used.

**Significantly associated features are capable of differentiating between 6' and 3' HA-sialoglycan interactions.** In the global specificity analysis above, when compared to background lectin-glycan interactions, HA interactions with 6' NeuAc-terminal glycans and 3' NeuAc-terminal glycans have very similar enrichment and depletion patterns matching those observed from the terminal NeuAc glycans (S12 Fig). However, direct, univariable statistical comparison of HA interactions with 6' NeuAc-terminal glycans against interactions with 3' NeuAc-terminal glycans with a weighted Wilcoxon-Mann-Whitney test revealed 35 features significantly associated with the presence of α2,6-NeuAc-terminal glycans (q < 0.01). To demonstrate that these significantly associated features captured determinants of fine HA specificity, unsupervised clustering using the correlations between the 35 significant features (Fig 6A,

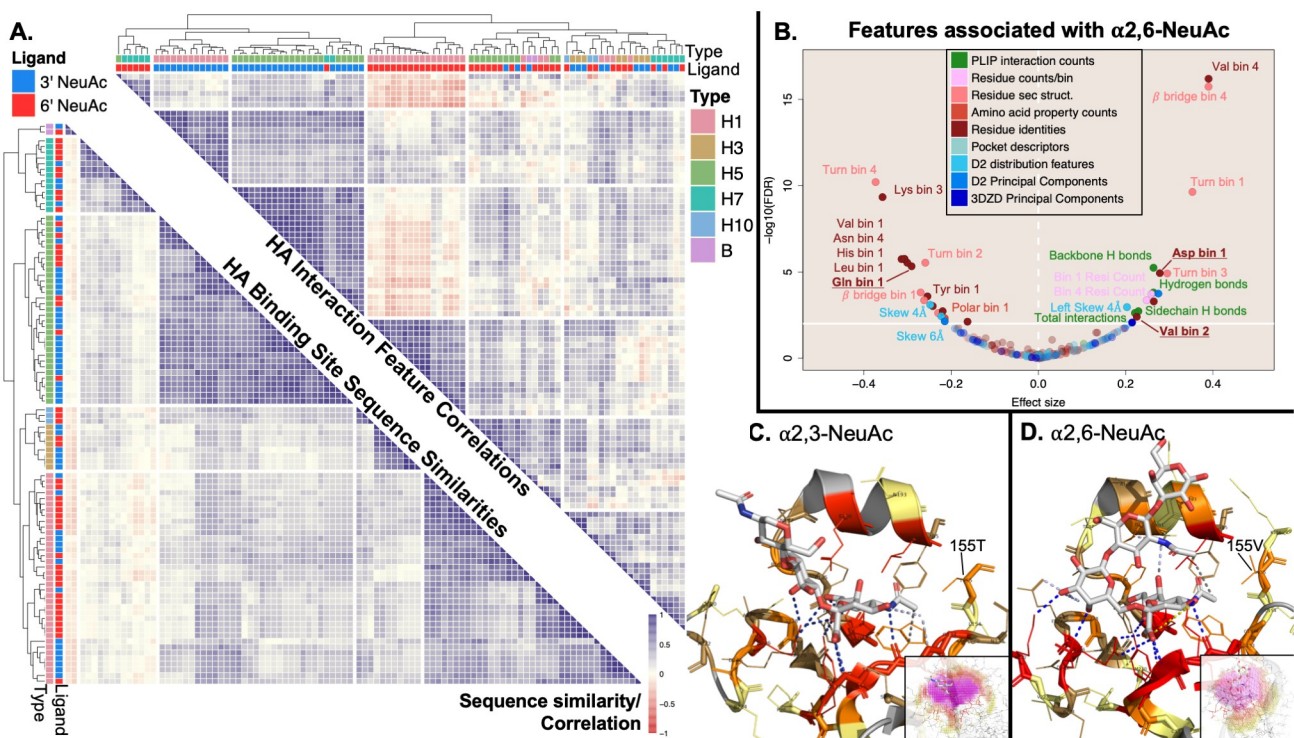

**Fig 6. Focused analysis of influenza HA binding sites reveals significant and discriminative features associated with binding of human-like sialoglycans over avian-like sialogylcans.** Clustering HA interactions by significant interaction features discriminates those recognizing 3' vs. 6' NeuAc terminal glycans, while clustering by interaction sequence simply recapitulates influenza strain. Panel A shows that clustering of the 96 HA-3'/6' αNeuAc interactions using correlations from the 35 significantly associated features allows for much cleaner grouping of interactions by ligand-type (upper-right-triangular similarity matrix) compared to interaction clustering using the alignment of the sequence of binding site residues leading to perfect grouping of interactions by influenza strain and HA subtype in the lower-left-triangular similarity matrix. Comparisons between hemagglutinin structures with 6' αNeuAc-terminal glycans versus 3' αNeuAc-terminal glycans reveal 35 features that are significantly associated with the presence of 6' αNeuAc-terminal glycans (panel B) displayed in the same manner as in Fig 2 with points discussed directly in the text bolded and underlined. These features are found in representative interaction structures between the respective glycans and HA proteins in panels C & D. In panel A, the upper-right-triangular matrix was constructed by calculating the pairwise Pearson correlations for all interactions using the scaled values of the 35 significant interaction features. The lower-left-triangular similarity matrix was constructed from sequence similarity scores using Needleman-Wunch to align binding site sequences with the BLOSUM62 substitution matrix. In panel B, significance and effect size were determined by a Wilcoxon-Mann-Whitney test weighted by influenza strain/hemagglutinin subtype and UniProt ID, with a significance threshold of q < 0.01 (solid horizontal white line) by the Benjamini-Hochberg Procedure. Panel C shows HA from H1N1 (Puerto Rico/8/1934) (dual specificity) complexed with an avian sialopentasccharide, although only the three terminal sugars were resolved (PDB ID: 1RVX). Panel D shows HA from H1N1 (California/4/09) in complex with a human sialopentasccharide (PDB ID: 3UBE). Both panels C and D use the same color scheme for lectins, PLIP interactions, and glycans as in Fig 5.

upper-right-triangular matrix) showed much clearer separation of interactions by glycan identity compared to clustering using similarity between the linear sequences of the binding site, which in fact led to perfect separation by influenza type/HA subtype instead of glycan identity (Fig 6A, lower-left-triangular matrix). The separation by ligand type while using the set of 35 significant features was especially clear within the H1, H5, and H7 subtypes. Unsupervised hierarchical clustering was used as an alternative means to demonstrate discriminative power of these 35 features since there were too few interaction examples to allow for rigorous cross-validation of a predictive classifier while controlling for homology among subtypes.

**Systematic characterization and comparison recovers known mutations driving HA specificity from 3' to 6' αNeuAc-terminal glycans.** The 35 significantly associated features (Fig 6B), primarily composed of residue-based features, highlight determinants of HA specificity for 6' αNeuAc-terminal glycans (S4 Table) over 3' αNeuAc-terminal glycans (S5 Table). The direction of association and significance for all 221 features can be found in S12 Fig. As a

visual aid to help interpret these features, representative interactions with each of these glycans are shown in Fig 6C and 6D. The NeuAc terminal glycans appeared in their typical conformations, with the 3' αNeuAc-terminal glycan oriented towards the 190-helix (Fig 6C) and the 6' αNeuAc-terminal glycan exiting over the 220-loop (Fig 6D). Representative interactions were selected as being the closest to the weighted feature-specific means for each glycan type and both interactions were with an HA from the H1 subtype.

The well-characterized E190D and G220D mutations in H1 subtypes are key mutations shifting HA specificity toward human-like glycans [54, 55] and were captured by a significant enrichment of aspartate in residues closest to the glycan, as well as both aspartate substitutions appearing in the the representative interaction with the 6' αNeuAc-terminal glycan but only G220D appearing in the representative interaction with the 3' αNeuAc-terminal glycan (Fig 6B–6D). Additionally, the primary mutations shifting specificity toward human-like sialic acid glycans in H2 and H3 subtypes are Q226L and G228S, with Q226L also playing a role in H7 subtypes and artificially induced Q226L/G228S substitutions in H5 decreased binding with 3' αNeuAc-terminal glycans [54, 56, 57]. The central substitution for these subtypes (Q226L) was captured in the significant depletion of glutamine in the closest bin to the glycan (Fig 6B). While not significant, glycine was depleted in the third bin, the bin in which 228G was found in interactions with H1 in Fig 6C and 6D, and serine was enriched in the second bin in closer contact with the glycan (S12 Fig). Thus the systematic, fine specificity analysis performed here successfully recaptured meaningful substitutions and mutations without prior knowledge even though they were present in a reduced number of HA subtypes. It should be noted that a number of features that were significantly associated with 6' αNeuAc terminal glycans resulted from the different conformations of the glycans seen in Fig 6C and 6D rather than specific mutations. This point is elaborated in section 3.

**Systematic characterization and comparison uncovers a potentially novel physicochemical determinant of 6' αNeuAc HA specificity.**   Valine was significantly enriched in bin 2 when 6' αNeuAc-terminal glycans were present in the interaction site (Fig 6B). Possible explanations from literature for this association include successive changes in Q226L→I→V in H3 HA leading to reduced binding to avian glycans [58] or G187V contributing to human sialogylcan binding preferences in H7 HA [57]. However, none of the interactions involving H3 HA contained any valine residues in bin 2, and the G187V substitution only placed valine into bin 2 for H7N9 HA interactions and it seems unlikely that this significance could be achieved by this single subtype, especially with two other H7Nx subtypes present without valine in bin 2 accounting for two-thirds of the weight attributed to the H7 subtype.

Notably, position 155 in the representative interaction with a 3' αNeuAc glycan was occupied by threonine (Fig 6C) while the representative interaction with a 6' αNeuAc glycan contained 155V (Fig 6D), with position 155 falling into the second bin for each interaction and contributing to valine enrichment in bin 2. Valine residues occurred at analogous positions in HA structures from H10 (146V) and from influenza B (160V) as seen in a multiple sequence alignment with sequences from each HA structure (S13 Fig). Of interest, subtype H10 HA has strong avidity for human sialoglycans but a stronger preference for avian sialoglycans [59] and influenza type B is typically found only in humans [60]. Within the available structures, only subtype H1 had examples with both valine and another amino acid (threonine) at that position. It appears that in H1 interactions with 3' αNeuAc glycans, threonine was oriented closer to the glycan compared to valine and was more likely to have a hydrophobic contact with the terminal carbon in the N-acetyl group of NeuAc. However, in H1 structures with 6' αNeuAc glycans, threonine could be found slightly further from the glycan and seemed less likely to have a hydrophobic contact compared to valine (S13 Fig). There were not enough interactions available for this comparison to give high confidence in this potential mechanism, but when

taken into consideration with the occurrence of valine at analogous positions in hemagglutinins from influenza types/subtypes with high avidity for human-like sialoglycans, T155V appears more likely to contribute towards the fine specificity of HA for 6' over 3' αNeuAc-terminal glycans, especially in the H1 subtype. To our knowledge, this position has not been previously noted to be associated with human-like versus avian-like sialoglycan specificity and more detailed analyses and experimental studies are warranted.

## 3 Discussion

Lectin-glycan interactions are critical in many natural and designed biological processes but we lack a detailed and comprehensive understanding of the features of these interactions responsible for their specificity. To this end, we have characterized geometric and physicochemical components of over 4,000 interactions, thus enabling systematic investigation of determinants of lectins' glycan-binding preferences and validation of these characterizations by assessment of their predictive power. Investigations into lectin specificity between all glycans (global specificity) as well as between slightly but impactfully altered glycans (fine specificity) recovered known similarities in glycan-binding preferences of lectins such as between mannose and glucose, sialic acid containing glycans, and lactose and LacNAc; highlighted less understood and less intuitive relationships in lectin-glycan interactions such as between Gal, GalNAc, Glc, and GlcNAc; and identified previously uncharacterized mutations potentially playing a role in influenza hemagglutinin glycan specificity, all while providing insights into potential mechanisms of specificity.

In contrast to previous efforts to probe general protein-glycan interaction structures, this work focused on lectins and the identification of binding site features conserved across non-homologous lectins with shared specificities, benefiting from the curated glycan identities contained within UniLectin3D. Our approach begins to provide insight into mechanistic determinants of lectin specificity and demonstrates that characteristics of lectin binding sites exist that can potentially be used to predict the specificity of novel, uncharacterized lectins. By demonstrating the feasibility of predicting lectin-glycan interactions from statistical associations with structural and biochemical features, we provide a proof-of-concept for future investigation into determinants of global and fine lectin specificity. Our demonstration of the study of fine specificity of influenza HA managed to uncover a previously unreported mutation seeming to play a role in HA glycan binding despite extensive previous research on HA specificity, indicating very strong discovery potential when applied to any number of less-studied cases. Highlighted mechanistic insights into lectin specificity could help inform lectin engineering efforts. Furthermore, in future work where lectin binding site representations might be more robust to variation in glycan size and orientation, similar approaches could be used to predict glycan-binding preferences of uncharacterized putative lectins. As such, this work represents a step towards unlocking a much broader diversity of lectin specificities for use in glycobiology research and clinical applications.

The primary limitation of our study was reliance on occurrence of glycans in lectin crystal structures as an indication of lectin specificity. Larger glycans are difficult to resolve in crystal structures but most glycan binding motifs are more complex than the mono- and disaccharides primarily considered here. In addition, overall glycan quality in the PDB is still relatively poor although efforts are ongoing to better annotate, curate, and catalog glycan structures in the PDB [61, 62]. Manual curation of glycan identities in the UniLectin3D database aid greatly in this regard, but there are still lingering discrepancies between the glycan IUPAC label and exact composition and modifications present on the observed structures that might add noise to observed associations. An additional limitation stemmed from our use of the individual

glycans to define each lectin-glycan interaction. While this approach is the most straightforward and our findings demonstrate its utility, it introduces additional variability and complicates the interpretation of results by introducing differences in the interaction characterization depending on the size and orientation of the glycan and separate from differences in the lectin binding site. Additionally, monosaccharides have more orientational freedom and can be found in conformations not realistically achieved in more biologically relevant polysaccharide contexts. This is reflected in a number of associated features from Fig 6B as well as the strong effect of glycan size on the 3D pocket features in Fig 4C.

Our approach uncovered realistic novel and confirmed determinants of lectin specificity while demonstrating the predictive potential of featurized lectin binding sites for study of lectins' glycan-binding preferences. Moving forward, the implementation of holistic lectin binding site definitions would enable more robust study of the specificity and promiscuity of lectin binding sites accommodating multiple glycan ligands. While we have focused strictly on pocket features, global lectin characteristics, e.g., fold information and valency, may also prove useful for future efforts focused on lectin specificity. Furthermore, incorporation of more informative and relevant studies of lectin specificity such as glycan microarray data will allow for more detailed studies of features determining lectin specificity, as well as the application of predictive models to unbound lectin structures or even homology-derived structures of putative lectin sequences.

## Methods

### Pre-processing

UniLectin3D data was shared by [39] on March 5th of 2020. After eliminating entries with no ligands, no accessible structural data, or resolution worse than 4 Å and manually annotating missing UniProt IDs for 14 entries, 1,376 entries of lectin-glycan structures remained, representing 412 unique lectins. All UniProtIDs, IUPAC glycan ligands, and associated UniLectin3D-provided information for these structures are available in S1 File. PDB files were accessed on August 28th, 2020 after the PDB carbohydrate remediation project. All protein and glycan structures were processed using BioPython v1.78 [63, 64] and visualized and rendered using PyMol v2.4.02 [65].

To identify glycan binding sites, each PDB file was processed with a forked version of the PLIP tool v1.4.5 [46] slightly modified to avoid excluding glycan ligands as artifacts and to facilitate downstream processing of results. All 9,828 PLIP-detected interactions were processed, cleaned, and excluded when necessary. Cleaning and exclusion involved merging PLIP interactions from separate components of the same glycan missing a glycosidic bond; capturing $Ca^{2+}$ ions in interactions; removing non-glycan ligands as determined using RDKit [66] from PDB-provided simplified molecular-input line-entry system (SMILES) representations when possible and otherwise using PLIP-determined SMILES representations; removing glycosylation occurrences within 1.7 Å of a serine, threonine, asparagine, tryptophan, or cysteine residue of the lectin; removing interactions with peptide chains used to display the glycan of interest, and excluding multiple copies of the same glycan in different anomeric conformations.

A final round of filtering was performed manually for 39 structures containing glycans with fewer than 3 PLIP interactions with the lectin or fewer than 3 residues within 4.5 Å of the protein, resulting in the removal of 44 interactions from 23 of the structures due to missed glycosylation occurrences, absent glycosidic bonds with reducing-terminal sugars, or non-specific interactions/crystallization artifacts. There remained 4,088 binding interactions from 1,364

PDB entries with high confidence of true lectin-glycan interactions, highlighting the absolute necessity of careful quality considerations in this sort of systematic structural analysis.

### PLIP feature generation

The PLIP interaction features were represented as counts of each type of interaction as well as the total number of glycan-lectin interactions. Interaction types included total hydrogen bonds, backbone hydrogen bonds, sidechain hydrogen bonds, water bridges, salt bridges (electrostatic interactions), hydrophobic contacts, halogen bonds, metal complexes, π-stacking, and π-cation interactions. These features are easily accessible to the community because the PLIP tool is freely available through a web server and the PLIP reports are included in the interactive UniLectin3D database, thereby decreasing technical barriers to usage.

### 3D pocket identification

Existing computational pocket-detection tools are primarily designed to identify protein binding sites suitable for small-molecule therapeutics [67], and we found that consequently none of the most commonly used tools were suitable for probing and characterizing the variable and often fairly shallow lectin binding sites [23]. Thus we employed an approach inspired by Vol-Site [68], placing voxels placed around the glycan to fill the relevant available space in the lectin binding site, but adapted to the unique geometries of lectins. A 3D grid was placed around the glycan ligand with points spaced approximately 0.79 Å apart on each coordinate plane, so that each point can be thought of as the center of a voxel with a volume of 0.5 $Å^3$. Points within 2.5 Å of a protein heavy atom were considered to be below the surface of the protein and were excluded, as were points that extended beyond the convex hull of the lectin binding site surface [69, 70]. This surface was found using the van der Waals surface of the lectin binding site as determined by PyMol [65]. Voxels were further limited by their distance from the closest glycan heavy atom to ensure the relevant region of the protein concavity was being captured, and to account for the higher variability in the shape, size, and extent of lectin binding sites, the threshold limiting that distance was set at four separate values for each interaction: 4, 6, 8, and 10 Å. This allowed for more information about the shape of the pocket to be extracted by considering how the characterization changes at each threshold. Finally, to ensure elimination of spurious voxels unlikely to represent 3D space potentially occupied by the glycan, the points were clustered with the density-based clustering algorithm DB-SCAN as implemented in scikit-learn v0.23.2 [71, 72]. The DB-SCAN neighborhood eps ($\epsilon$) was set to include the 26 points that could potentially directly neighbor a given point, and at least 50% of those points were required to be present (MinPnts = 13) to consider a point as a core point. Points were excluded if they were labelled as noise by DB-SCAN or if their assigned cluster had less than 15 $Å^3$ of combined volume or fewer than 10% of its points within 2 Å of a heavy glycan atom, ensuring the space being characterized was relevant and accessible to the glycan. An exception was made for clusters that passed the 10% threshold in pocket representations generated with a lower distance-from-the-glycan threshold but fell below 10% at larger thresholds. This approach characterized 4,074 of the 4,088 interaction pockets and the remaining 14 interactions sites were flat or convex with no pocket to find.

### 3D pocket feature generation

For each distance threshold used to generate the voxelized representation of the pocket, the general pocket descriptor features included the volume of the pocket and percent of voxels found on the surface of the pocket. Voxels were labelled as either "buried" or "surface" voxels by the number of the 26 possible directly-neighboring voxels present, where surface voxels had

fewer than 23 of these 26 present. This value represented a compromise between an overly restrictive definition for buried voxels such that pockets below a certain size would be comprised entirely of surface voxels, and an overly loose definition that would neglect some of the surface. The voxelized pocket representations were further characterized by two robust, rotationally-invariant, and complimentary approaches: the D2 distributions which represent information about pocket shape while being heavily influenced by the extent and dimensions of the pocket, and the 3D Zernike descriptors (3DZDs) which succinctly represent the pocket shapes without influence from the overall size of the pocket.

D2 distributions were found by computing all pairwise distances between the centroids of surface voxels [49] and features for each threshold were derived from the statistics describing this distribution when placed into 0.5 Å bins. Distribution statistics included variance, 1st quartile, median, 3rd quartile, left & right skew, and the number of major and minor local maxima found after smoothing the distribution with a moving 9-point and 5-point average respectively. To somewhat reduce the influence of overall pocket size and allow pocket shape to carry more weight in features, the pairwise distance measures were transformed for each interaction and each pocket threshold by fitting the measures into 40 equal-sized bins scaled to the maximum observed distance in the pocket, calculating the frequency of measures in each bin, and concatenating the resultant vector from each threshold into one 160-dimensional vector for each interaction. The top 20 principal components (PCs) describing these 160-dimensional vectors (accounting for approximately 80% of the total variance) were used as the D2 principal component features.

To represent the diversity of possible pocket shapes independently from pocket size, rotationally-invariant 3DZDs based on 3D Zernike moments [48], computed up to the 10th order, were determined for point clouds defined by the voxels' centroids, resulting in 36 3DZDs per pocket, concatenated into a single 144-dimensional vector over the different thresholds. These 3DZDs were found using the software from Daberdaku and Ferrari, [47] modified to accept the point cloud representations without additional annotation. Point cloud processing to transform and scale coordinates within the unit sphere was inspired by Grandison et al. [73]. Compared to previous studies, the use of 10th order 3DZDs appeared to be sufficient to capture the shape of pockets which are already somewhat smoothed in the voxelization process [74]. The top 17 principal components (PCs) describing the 144 concatenated 3DZDs from each distance threshold for each interaction site were used as the 3DZD principal component features. These top 17 PCs accounted for approximately 80% of the total variance.

## Binding site residue characteristics

To allow for more complete and continuous lectin binding sites, the binding sites were expanded from PLIP-defined binding residues to include two residues on each side of an interacting residue as well as residues immediately between two binding site residues. Secondary structure, backbone angles, and solvent accessibility information for binding site residues were calculated from the structure files using DSSP v2.3.0 [75, 76].

Binding site residues were binned by distance from the glycan to mitigate the high likelihood that the observed interacting residues in the crystal structure are not the only residues contributing to interactions stemming from the flexibility of proteins and especially glycans, the potential for the solved conformation to be influenced by artifacts in the crystallization process, and the high probability of the existence of alternate low-energy conformations. These bins served as a rough approximation of the probability of interacting with the glycan and the overall physicochemical environment of the binding site. Features from the binding site were generated by placing each residue within 8 Å of the glycan into four bins, $\leq 3.5$Å,

3.5 – 4.5Å, 4.5 – 6.5Å, and 6.5 – 8Å, by the shortest distance between a residue-glycan heavy atom pair. For each bin, features included the total number of residues, the frequency of each of the 20 common amino acids, the frequencies of the 7 DSSP-defined secondary structures observed in lectin binding sites (α-helix, β-bridge, β-strand, $3_{10}$-helix, hydrogen bonded turns, and loops/irregular structure), and the frequencies of 5 physicochemical classes of amino acids (nonpolar residues Gly, Ala, Val, Leu, Ile, Met, & Mse; polar residues Ser, Thr, Cys, Pro, Asn, & Gln; positively charged residues Lys, Arg, & His; negatively charged residues Asp & Glu; and aromatic residues Phe, Tyr, & Trp). One additional feature was included in the first bin to store the number of $Ca^{2+}$ ions present within 3 Å of the glycan [77].

## Homology between lectins

To generate consistent lectin sequences from each structure, the protein sequences from each chain were clustered with CD-HIT [78, 79] at 90% identity and non-redundant lectin sequences from each structure were constructed using the representative sequence of each cluster ordered by the lowest chain ID from each corresponding cluster. The non-redundant lectin sequences were clustered again with CD-HIT at 50% sequence identity to obtain the 225 clusters of homologous lectins.

## One-versus-all statistical associations

Featurized lectin-glycan interactions were compared using weighted Wilcoxon-Mann-Whitney (WMW) tests [50], a non-parametric approach to test whether the random samples from two different groups were sampled from the same underlying distribution. The WMW test is particularly useful here because it is well-suited for the ordinal and non-continuous variables present in our features, it works well with smaller sample sizes that are encountered among the less frequent glycans, and it does not require assumptions of normality which would not likely be met for many of the features [80]. Weights for the interactions control for redundancy from homologous lectins as well as repeated interactions from the same lectin. The total weight, i.e., the number of interactions, was equally divided among the clusters of homologous lectins. Each unique lectin in a cluster as determined by UniProt IDs was then allotted an equal proportion of the total cluster weight, and each observed interaction involving a unique lectin was assigned an equal proportion of the weight for the given lectin. Glycan symbols supplementing glycan names in figures follow the SNFG (Symbol Nomenclature for Glycans) system [81] as generated by [82].

The level of significance provided by the WMW test indicates the probability that the null hypothesis ($H_0$: $P(X < Y) = P(Y < X)$) where $X$ is a randomly sampled value for a given feature from the interactions containing a glycan of interest and $Y$ is a randomly sampled value from the interactions with all other glycans. Rejection of the null hypothesis allows for the acceptance of the alternative hypothesis ($H_1$: $P(X < Y) \neq P(Y < X)$) that the lectin interactions with the glycan of interest are enriched or depleted for the given feature compared to background interactions. The reported common language effect size can be interpreted as $P(Y < X) - 50\%$, or the probability that for a randomly sampled pair of feature values from the interactions with the glycan of interest and from all other glycan interactions, the value will be greater from the interaction with the glycan of interest [83].

To control for multiple hypothesis testing, Benjamini-Hochberg correction was applied for each glycan of interest [84] and the significance threshold was set to q < 0.01 to provide an FDR of 1%. Significance values of q < $10^{-16}$ were considered extremely significant such that further increased significance is not meaningfully interpretable, so to improve visualization in

Fig 2 these values were replaced with a random significance value sampled from a log-uniform distribution between $1 \times 10^{-16}$ and $3 \times 10^{-19}$.

## One-versus-all predictive modeling

Random forest models for each glycan of interest were trained using a leave-one-out cross-validation strategy such that models were trained on interactions from all but one of the 225 groups of homologous lectins and then used to predict the presence of the glycan in the withheld interactions. This process was repeated for each cluster of lectins that contained any interactions with the glycan of interest, training on all other interactions and testing on the withheld interactions to assess cross-validated performance. To balance the number of negative and positive examples, the majority class (negative examples) was downsampled during training. For each glycan-specific model, 2,000 trees were built at each iteration of the cross-validation. Within the training step, 5-fold cross-validation was used to aid in selecting the number of features to include (*mtry*) and assess training performance in a nested cross-validation approach [85]. The number of features tried from the 221 total features was considered from a range around the default of $\sqrt{221}: \left[\sqrt{221} - \frac{\sqrt{221}}{2}, \sqrt{221} + \frac{\sqrt{221}}{2}\right]$. Random forest and cross validation were performed with randomForest v4.16-14 and caret v6.0-86 [86, 87].

Since, as discussed, negative labels are not always meaningful in this data (perhaps that exact structure has not yet been solved) and recall and precision do not rely on the true negative rate, training performance was assessed using the harmonic mean of recall and precision known as the $F_\beta$ score where $\beta$ was set to 2, placing more weight on recall performance ($r$) compared to precision ($p$) as recall only uses positively labelled data.

$$F_\beta = (1 + \beta^2) \frac{pr}{\beta^2 p + r}$$

In order to further control for redundant lectin interactions beyond the level of homology clustering at both the training and validation steps, a set of dissimilar positive interactions was randomly sampled from interactions within each group of similar lectins, followed by a set of negative interactions dissimilar from each other (as well as the initially sampled positive interactions). Diverse interactions were assessed by Euclidean distance between vectors comprised of their features each scaled between 0 and 1. Once an interaction was sampled, all remaining interactions within a thresholded distance were excluded, and sampling continued as long as eligible interactions remained. The threshold used was equivalent to the median of all pairwise Euclidean distances between interactions. Distances were found using philentropy v0.4.0 [88].

To account for variation in the random sampling of test cases for each validation step of this approach, the sampling and prediction was repeated 10 times at each iteration of the leave-one-out cross-validation. To additionally account for variation in the training data and the stochasticity of the RFs, the leave-one-out cross-validation approach was repeated 10 times for each glycan of interest. As a result, 100 samples of RF performance were measured for each glycan-specific RF classifier and displayed in Fig 3. The "null model" random classifier was built, trained, and tested exactly the same way except the labels for the interactions for each glycan of interest were shuffled at the beginning of each of the 10 repeats of the leave-one-out training/testing procedure.

Feature importance was determined from the recorded mean decrease in Gini impurity for the features from each model built in each iteration of the validation procedure. To compartmentalize complications in interpreting feature importance arising from correlated features describing the size of the interaction pocket [89], features were stratified into the 3 categories outlined in Fig 1A–1D before ranking feature importances from each model, and the median

rank importance across each repeat of each round of leave-one-out cross-validation was computed for features within their respective categories. Features whose median stratified importance rank was at least in the 75th percentile were considered to be highly predictive.

## Glycan recognition similarity analysis

To identify similar determinants of specificity in lectins recognizing different glycans, the common effect size values of the features from the weighted WMW tests for each glycan of interest were correlated using Pearson correlation and the glycans were hierarchically clustered with complete linkage using correlation as the distance metric. Heatmaps were generated using pheatmap v1.0.12 [90].

To find representative interaction examples displayed in Fig 5A–5C, weighted average feature values with features scaled between 0 & 1 from interactions with each glycan were calculated using the same interaction weights applied in the weighted WMW tests. Selected representative interactions for each glycan had the shortest Euclidean distance between its scaled features and the glycan-specific weighted feature averages. Representative interactions from the fine specificity of influenza HA displayed in Fig 6C and 6D were found in the same manner, except only the 35 significantly associated features were used due to the strong similarities between the interactions being compared.

## Determinants of fine specificity for αNeuAc glycans

Limiting lectins to influenza HA proteins focused investigation on a shared, conserved binding site, but homology between viral subtypes still could bias the analysis. A weighted WMW test [50] was used to compare features from these 96 HA-α2,3/6-NeuAc terminal glycan interactions (47 from 6' NeuAc & 49 from 3' NeuAc) following the same approach as used to investigate one-vs-all statistical associations, except that the total weight was first divided between the 6 influenza type/HA subtypes present (Influenza B & influenza A H1, H3, H5, H7, & H10) instead of relying on the clusters of homologous lectins. Of note: 105 HA interactions were initially identified with these glycans, but 9 were observed to be cases of missed glycosylation or non-specific interaction and were excluded from this fine specificity analysis.

Hierarchical clustering in Fig 6A was performed with complete linkage. Sequence similarity scores used to cluster interactions from the binding site sequences were calculated using Needleman-Wunch alignment and BLOSUM62 substitution matrix [91, 92], implemented in protr v1.6–2 with default parameters [93]. The binding site sequences from the HA structures were found using all residues identified as being contained within the binding site as defined by the expanded PLIP binding site residues described previously ("Binding site residue characteristics"), grouped in order by chain ID and residue number. To cluster the interactions based on the interaction features, the similarity scores were found by calculating pairwise Pearson correlations of the 35 significant interactions features scaled between 0 and 1. Pearson correlation of the scaled features was used over Spearman correlation of the raw features due to the large variation in range present in the feature set not allowing for informative ranking of all features.

The multiple sequence alignment in S13 Fig used the non-redundant sequences extracted for homology comparison and was performed and rendered in Seaview v5.0 with Clustal Omega and default parameters [94, 95]. Hierarchical clustering in S13 Fig was performed with complete linkage based on pairwise global alignment as performed for the HA binding site sequence alignments.

## Supporting information

**S1 File. Lectin binding sites with features and UniLectin3D information.** The curated set of 4,088 lectin-glycan interactions with associated UniLectin3D information and the values of the 221 features used in this analysis.
(CSV)

**S1 Fig. Non-redundant lectin sequence extraction.** Schematic of the workflow used to extract non-redundant sequences from structure files.
(TIF)

**S2 Fig. Histogram of the counts of unique lectins within each homology cluster.** The number of unique lectins (as defined by UniProtID) within each homology cluster generated with CD-HIT at 50% sequence identity. Most homology clusters only contained 5 or fewer unique lectins, but some very well studied lectins and homologous lectins were grouped into very large homology clusters.
(TIF)

**S3 Fig. Occurrence of each unique glycan ligand across groups of non-homologous lectins.** Frequencies of all 226 unique IUPAC-labelled glycans within each cluster of homologous lectins. The top 12 individual glycans (vertical line) each appeared in complex with at least 5% of the 225 clusters of homologous lectins (horizontal line). Information about each glycan is provided below each bar of the barplot, including membership of one of the three groups of glycans (terminal NeuAc, high mannose, and terminal fucose).
(TIF)

**S4 Fig. Occurrences of the 15 glycans of interest across groups of non-homologous lectins.** Panel A shows the same values as S3 Fig for the 12 most commonly bound glycans (right of the dotted vertical line), annotated with their corresponding IUPAC names and SNFG symbols, as well as the frequencies of the 3 groups of glycans (left of the dotted vertical line) appearing bound to any lectins in the 225 homology clusters with their representative SNFG symbols. Panel B shows the actual distributions of training samples (on a log scale) used for each individual RF model for each glycan at reach repeat and interaction of the leave-one-out cross-validation. These distributions appeared fairly proportional to the relative frequencies of each glycan in panel A.
(TIF)

**S5 Fig. Glycan-specific RF model training performances.** Training performance of glycan-specific RF models measured with nested 5x cross-validation. Recall (left y-axis) and precision (right y-axis) of glycan-specific random forest models is shown by the split violin plots, with the left-hand distributions depicting recall and the right-hand distributions depicting precision. The pairs of notch boxplots for each glycan show the performance of the random classifiers trained on data with shuffled labels, where again the left-hand boxplots depict the random classifiers' recall and the right-hand boxplots depict their precision.
(TIF)

**S6 Fig. Glycan-specific RF model training performance is correlated with the number of training examples.** Training performance of glycan-specific RF models summarized by $F_2$ scores combining recall and precision with greater emphasis on recall, plotted against the number of samples used in training each specific model. Glycan labels are placed on the mean $F_2$ and sample numbers for each glycan. Training nested-cross-validation performance is fairly correlated with the number of samples available for training (Pearson correlation $\rho = 0.39$, $p < 0.001$).
(TIF)

**S7 Fig. Median feature importance percentiles from glycan-specific RF models.** Median feature importance percentiles from each glycan-specific RF model determined via mean decrease in Gini impurity. Size-correlated pocket features (blue) were often grouped together at a higher importance level, motivating the stratification by feature type to prevent multicollinearity from one feature type affecting other features. Points were colored with the same color scheme detailed in Figs 1, 2, 4 and 6.
(TIF)

**S8 Fig. Glycan-specific median RF feature importance percentiles within residue features.** Median feature importance percentiles of residue-based features (within the residue features only) from each glycan-specific RF model determined via mean decrease in Gini impurity. Features with median importance in at least the 75th percentile (horizontal line) were considered highly predictive. Points were colored with the same color scheme detailed in Figs 1, 2, 4 and 6.
(TIF)

**S9 Fig. Glycan-specific median RF feature importance percentiles within pocket features.** Median feature importance percentiles of pocket-based features (within the pocket features only) from each glycan-specific RF model determined via mean decrease in Gini impurity. Features with median importance in at least the 75th percentile (horizontal line) were considered highly predictive. Points were colored with the same color scheme detailed in Figs 1, 2, 4 and 6.
(TIF)

**S10 Fig. Glycan-specific median RF feature importance percentiles within PLIP features.** Median feature importance percentiles of PLIP features (within the PLIP features only) from each glycan-specific RF model determined via mean decrease in Gini impurity. Features with median importance in at least the 75th percentile (horizontal line) were considered highly predictive. Points were colored with the same color scheme detailed in Figs 1, 2, 4 and 6.
(TIF)

**S11 Fig. Feature-type stratified percentiles of RF feature importances are generally correlated with the degree of feature enrichment from the WMW test.** Median ranked feature importance percentiles (stratified by feature type) are plotted together against the absolute value of the weighted WMW effect size. In general, the stronger the observed association from the WMW test for a given feature, the more likely the feature was to be highly predictive. The dotted horizontal line indicates the 75th percentile threshold. Points that are bolded represent features that passed the 75th percentile for feature importance and were found to signficant from the weighted WMW test at q < 0.01 following the Benjamini-Hochberg procedure. Points were colored with the same color scheme detailed in Figs 1, 2, 4 and 6.
(TIF)

**S12 Fig. Features associated with 6' vs 3' sialoglycans and compared to observed associations from NeuAc glycans vs all other glycans.** Enrichment and depletion patterns in the 221 features for 6' NeuAc glycans compared to 3' NeuAc glycans (bottom row) determined by a weighted WMW test. Bullet points indicate q < 0.01 by Benjamini-Hochberg correction. The first row shows the associations for terminal NeuAc glycans compared to background from Fig 4 for ease of comparison, and the second and third row show comparisons of 6' NeuAc glycans and 3' NeuAc glycans in HA binding sites compared to background interactions.
(TIF)

**S13 Fig. Valine at position 155 in H1 appears to be a previously uncharacterized mutation associated with 6' sialoglycan specificity.** Valine appears at position 155 (H1 numbering) in

H1, H10, and type B hemagglutinin structures, as shown by the multiple sequence alignment (Clustal Omega), visualized with Seaview, and clustered by global sequence identity (BLOSUM 62), as show in panel A. Panel B shows the distributions of the measured minimum distance from any atom in the residue at position 155 to the closest heavy glycan atom (usually the terminal carbon of the N-acetyl group) within all HA structures from H1N1. When complexed with 3' sialoglycans, threonine is usually oriented closer to the glycan compared to valine, and has a hydrophobic contact with the sugar in two of the structures (compared to one structure when valine is present). When complexed with 6' sialyoglycans, valine is more tightly grouped closer to the glycan and has one observed hydrophobic interaction with the glycan while threonine has no contacts.
(TIF)

**S1 Table. UniLectin3D-assigned IUPAC glycan names within the terminal NeuAc group of glycans.**
(PDF)

**S2 Table. UniLectin3D-assigned IUPAC glycan names within the high mannose group of glycans.**
(PDF)

**S3 Table. UniLectin3D-assigned IUPAC glycan names within the terminal fucose group of glycans.**
(PDF)

**S4 Table. UniLectin3D-assigned IUPAC glycan names within the 6' αNeuAc-terminal glycans complexed with influenza hemagglutinin.**
(PDF)

**S5 Table. UniLectin3D-assigned IUPAC glycan names within the 3' αNeuAc-terminal glycans complexed with influenza hemagglutinin.**
(PDF)

## Acknowledgments

We would like to thank Dr. François Bonnardel for sharing bulk data from UniLectin3D; Dr. Sebastian Daberdaku for sharing his source code used to generate the 3DZDs; Bowen Dai for his contribution to processing binding site surfaces for the determination of the convex hull; and Professors Margaret Ackerman, Karl Griswold, and Jiwon Lee of the Dartmouth Thayer School of Engineering, as well as their lab members, for their continued feedback on this project and many others at all stages of development.

## Author Contributions

**Conceptualization:** Daniel E. Mattox, Chris Bailey-Kellogg.

**Data curation:** Daniel E. Mattox.

**Formal analysis:** Daniel E. Mattox.

**Funding acquisition:** Daniel E. Mattox, Chris Bailey-Kellogg.

**Investigation:** Daniel E. Mattox, Chris Bailey-Kellogg.

**Methodology:** Daniel E. Mattox.

**Software:** Daniel E. Mattox.

**Supervision:** Chris Bailey-Kellogg.

**Visualization:** Daniel E. Mattox.

**Writing – original draft:** Daniel E. Mattox.

**Writing – review & editing:** Daniel E. Mattox, Chris Bailey-Kellogg.

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
