## [Decision Letter · Decision Letter 0]

26 Jul 2021

Dear Dr. Bailey-Kellogg,

Thank you very much for submitting your manuscript "Comprehensive analysis of lectin-glycan interactions reveals determinants of lectin specificity" for consideration at PLOS Computational Biology.

As with all papers reviewed by the journal, your manuscript was reviewed by members of the editorial board and by several independent reviewers. In light of the reviews (below this email), we would like to invite the resubmission of a significantly-revised version that takes into account the reviewers' comments.

We cannot make any decision about publication until we have seen the revised manuscript and your response to the reviewers' comments. Your revised manuscript is also likely to be sent to reviewers for further evaluation.

Sincerely,

Rebecca C. Wade

Associate Editor

PLOS Computational Biology

Arne Elofsson

Deputy Editor

PLOS Computational Biology

Reviewer's Responses to Questions

**Comments to the Authors:**

Reviewer #1: In this manuscript the authors present an extensive statistical classification of lectins from the UniLectin database with the aim of determining the fundamental structural and electrostatic/hydrophobic characteristics driving molecular recognition of specific glycan epitopes and monosaccharides. This classification is based on 221 features, the authors carefully selected as essential determinants driving binding. In my opinion this work is timely, and the aims address a very important matter, which solution would indeed open the field for the use of lectins in carbohydrate sequence, structure, and binding characterization. For this reason, I’d like to congratulate the authors for embarking in such interesting and potentially critical study for the advancement of glycoscience, and I hope the suggestion below will help improve not only the readability of the manuscript itself, but also (and more importantly) the predictive potentials of the method.

Indeed, after carefully reading the manuscript, I unfortunately did not find the results convincing, especially in terms of the ability of the method to predict the binding specificity of lectins. More specifically, the attribution of enriched and depleted features, which should be a discriminant for glycan specificity is unclear and in my opinion, heavily biased throughout by the prior knowledge of the preferential lectins’ binding epitope, so extremely hard to generalize for a blind prediction. As a demonstration of that, in my view the test case on HA illustrates a problem, as the HA binding site does not satisfy the characteristics the authors have highlighted as distinctive for a terminal sialic acid specific site, namely there is no high density of positively charged (at pH 7) residues, but actually an increase in negatively charged residues associated to the change in specificity from alpha(2-6/3), and the binding site is structurally quite shallow (no pockets). Therefore, the method may not identify the latter as a sialic acid binding site.

I believe that the problem may rest with the choice of glycans ‘categories’ used to train the method. For example, terminal sialic acids in the same group are either 2-3/6, there are also polysialic motifs with a terminal 2-8, also in the same group. Some of those sialylated glycans also contain fucose in the arms (LeX) and core fucose. Within this context it’s important to underline that terminal Sia in complex N-glycans is always linked to a LaNAc or to other structures in other glycans, so that it never comes alone and the binding site has evolved to form specific interactions along the disaccharide/trisaccharide 3D motif. Therefore, it is not reasonable to expect different features to play a role in selecting either terminal sialic acid or LacNAc, as both may be bound in the same site. Also, the generalization to terminal fucose is a bit ambiguous, as most of the chosen epitopes all contain 1-2 fucose, not terminal fucose in the arms (LeX) nor core fucose, which are (strictly speaking) both ‘terminal’ as nothing is linked to them.

As a suggestion that the authors may find useful, in my opionion the algorithm could benefit from a better-refined choice of the glycans substrates, so that they constitute uniform groups, consistent in terms of sequence and structure. Both these characteristics highly affect the 3D structure of the epitope, making it unique, which is fundamental as the lectin binding site specificity hinges on a structural and electrostatic complementarity to the epitope, that ultimately allows for recognition and binding. For example, Sia(2-3)LacNAc has a different structure relative to Sia(2-6)LacNAc, as it can be seen in the figures illustrating the HA test case; some lectins do bind either, but a general preference or promiscuity can indeed be revealed by a method trained to select for each of those separately, which would be incredibly powerful. Moreover, in this matter, I found unclear how a method not trained to distinguish 2-3 from 2-6 sialic acid can indeed predict changes in specificity in HA, and especially that an increase in negatively charged residues (which as mentioned earlier, goes against the described/reported characteristics of a terminal sialic acid-specific binding site) is associated to 2-3 preference.

Another question I could not find the answer to and that the authors may consider addressing/clarifying is, how was the specificity to isolated monosaccharides assessed? I am not aware of lectins binding isolated monosaccharides, such as fucose, sialic acid, i.e. not in a context of a larger glycan.

As a few minor points, I found the manuscript a bit too long (50 odd pages) and discursive. In my opinion, most of the method details could easily go into supplementary material, where the interested reader can find them, without disrupting the flow. Some of the graphs and figures are too difficult to read as the text is very small, e.g. Fig 4 and 6A. An additional feature not considered and that I believe may be useful is the lectin’s fold. Line 398, consider replacing with N-Acetyl Glucosamine and Galactosamine or GlcNAc and GalNAc.

Reviewer #2: The manuscript describes a novel bioinformatics analysis of 3D structures of lectin binding sites, with the aim of deciphering the structural basis of specificity and to obtain a predictive tool. The work is clearly described and the results are of high interest since clear separation could be obtained between different classes of specificity. Application to the fine specificity of hemagglutinins towards different sialylated oligosaccharides reached the prediction level. Altogether, it is a very original approach that will benefit to the glycobiology community.

Some points need to be clarified, or revised, in relation with the complex behaviour of lectins that may complicate some of the interpretation.

Main points

- Lectins recognize spatial arrangement of hydroxyl groups (and N-acetyl or methyl) that can be generated by different scaffold in a rather similar way. NeuAc and GlcNAc can be recognized by same lectin (WGA, PVL …) with different orientation of the ring. Same with Fuc and Man (LecB)… Did the authors considered crystal structures of one lectins with different ligand when appropriate ?

- The discussion about differences between mono/disaccharides and larger glycans bottom of page 22 is problematic. First, the term “conformational freedom” of monosaccharides should be replaced by “orientational freedom” . Second, the function of lectins is not to bind monosaccharides (opposite to transport protein) but to attach to complex glycan on surfaces. The fact that many are co-crystallized with monosaccharide is a bias from experiments (easier to obtain crystals). So the differences should be rather analysed in term of binding to branch structures (lewis, oligomann ) versus linear ones (sialylated)

- The separation between terminal sialylated and terminal fucose (Table S1 and S3 ) poses the problem of Sialyl Lexis x (NeuAc(a2-3)Gal(b1-4)[Fuc(a1-3)]GlcNAc that is listed as “terminal NeuAc” and not as “terminal Fuc” , while it can be bound by the Fuc moiety in some lectins

- The terminal Fuc compounds in Table 3 does not include Lewis x Gal(b1-4)[Fuc(a1-3)]GlcNAc that is a very common ligand in lectins / this should be checked

- As pointed by the authors, there are some errors in Unilectin3D, and it would be better not to reproduce there in the present article. Ligands 12, 16 and 37 of Table 1 do not exist (likely typing error) and should not be included (corrections has been requested in unilectin3D, so likely to be corrected now). Also not sure about meaning of ligand 33

Details to be corrected

Authors summary

Line 10 : glycans are attached to proteins and lipids

Line 12 : should be “influenza virus”

Line 14 : “sweeter” is catchy but not appropriate : higher affinity ?

Introduction :

Line 24 ; :lectins are proteins, not protein domains (they have carbohydrate binding domain and other ones)

Line 29 : hemagglutinin lectin : “lectin” should be removed

Legend of Figure 1 : The complex selected for illustration does not include “bacterial lipopolysaccharide” but only a disaccharide fragment. The E plot needs some more description . What is the y-axis (definition, values), what is the color coding of the x-axis ?

Reviewer #3: The manuscript “Comprehensive analysis of lectin-glycan interactions reveals determinants of lectin specificity” aims to develop a systematic study to highlight complementary physiochemical and geometric features which allow to define the specificity of the lectin-glycans binding. After a general overview about the state of the art and considering the advantages and limitations of the experimental and computational techniques used in the field, the authors introduce the open points that they aim to address in this paper.

In this study, the authors screen and choose over 1300 structures, representing more than 400 lectins in complex with 226 glycan ligands, clustered at 50% of identity. The resulting 225 clusters were in subjected to interaction weighting and sampling based at each step of the analysis to prevent disproportionated impact from better-represented lectins clusters. To define the specific features defining the lectin-glycans binding, the authors adopt the univariable comparative analysis with weighted Wilcoxon-Mann-Whitney test which reveals specific lectin binding pocket features, and the multivariable modeling with random forests in combination with the previous analysis which demonstrate that in specific cases (i.e. NeuAc terminal glycans, mannose monosaccharides and fucose monosaccharide) particular features combinations suffice to predict specific recognition. By integrating the 221 features as well as different type of features (i.e. pocket features, physicochemical environments and recognition motifs, 3D geometry based relationship, similarities in PLIP-characterizes atomic interaction) identified from the comparative and predictive approaches, the authors extracted global determinants of specificity that can be significant and predictive to identify the binding of similar glycans. Finally, the systematic analysis developed is adopted to successfully characterize the specific features which define the binding specificity between 3’ and 6’ α-NeuAc terminal glycans. Interestingly, the analysis not only recovers known mutations which drive the specificity of the above-mentioned binding but also uncovers a new potential physicochemical determinant for the 6’ α-NeuAc terminal glycan specificity.

Overall, the work is carefully carried out and clearly described. It represents a useful systematic study that increases knowledge of lectin-glycan binding features laying the basis to obtain further insight into the field.

I however suggest the following minor edits about the study:

- Figure 1. In panel B, the code-colour should be the same as for the other panels. In panel C, clarify the role of the additional components. Overall, I suggest to make the workflow of the analysis easier to handle by clarifying that the data in Fig.1 E contains the features from Fig.1 B-C-D. In addition, it is not clear to me why the legend in Fig.1E (squares under the histograms) does not follow the same order of the Figures (B-C-D in figures VS B-D-C in legend).

- Line 200. Is it glucose or galactose? It is not clear to me why the authors choose the glucose because, in my opinion, the trend of the galactose seems more similar with the other mentioned. Can you explain it?

- Line 204. As previous, in my opinion LacNAc seems to have a similar trend as NAcetylneuraminic acid and 3’ sialyllactose. Can you clarify it?

- Figure 6B. I would suggest to help the reader following the discussion in association with the figure by highlighting the point discussed with arrows or something like that.

**Have the authors made all data and (if applicable) computational code underlying the findings in their manuscript fully available?**

Reviewer #1: Yes

Reviewer #2: None

Reviewer #3: Yes

PLOS authors have the option to publish the peer review history of their article (what does this mean?). If published, this will include your full peer review and any attached files.

Reviewer #1: **Yes: **Elisa Fadda

Reviewer #2: No

Reviewer #3: **Yes: **Giulia Paiardi
---

## [Decision Letter · Decision Letter 1]

15 Sep 2021

Dear Dr. Bailey-Kellogg,

Thank you very much for submitting your manuscript "Comprehensive analysis of lectin-glycan interactions reveals determinants of lectin specificity" for consideration at PLOS Computational Biology. As with all papers reviewed by the journal, your manuscript was reviewed by members of the editorial board and by several independent reviewers. The reviewers appreciated the attention to an important topic. Based on the reviews, we are likely to accept this manuscript for publication, providing that you modify the manuscript according to the review recommendations.

Sincerely,

Rebecca C. Wade

Associate Editor

PLOS Computational Biology

Arne Elofsson

Deputy Editor

PLOS Computational Biology

[LINK]

Reviewer's Responses to Questions

**Comments to the Authors:**

Reviewer #1: I’d like to thank the authors for their answer to my questions and for the modifications they made to this version of the manuscript, which in my opinion is a strong improvement over the original submission. I particularly appreciated the fact that the authors explained much better to the reader the limitations of their method (see more below), which stems from the (relatively low) number, diversity and degree of completeness of the structural data available to date in UniLectin. Also, the HA case is not framed anymore as a proof of concept, at least in my opinion it isn’t, yet as a useful test of how looking at the selected features within the framework of a specific comparison between highly similar lectins, can indicate epitope specificity. I find this information extremely valuable to the whole community.

I have only minor suggestions to this version that I’m hoping the authors will consider taking on board, as those will take very little time and effort, hopefully.

The comments and points about the limitation of the method are now made clear in the Discussion section, so reaching the “really” interested readers, i.e. the one who go through all the results. Tp the broader readership’s benefit, I would suggest to add a sentence to that effect in the abstract.

Also, I find it would be important to indicate quantitatively what is the threshold for “statistical significance” used in the data presented in Figure 2, see caption. The authors could also refer the reader to the method section, where those thresholds could be discussed in detail, if needed.

I again recommend for the method to appear in the SI, or to shorten the manuscript some other way, yet the latter may be more time consuming.

Figure 6 panel A is unreadable, I understand perfectly well that those details/labels cannot be made larger, but if they can’t be read, they are hardly useful, in my opinion. That panel can maybe be transferred to SI and blown up?

Reviewer #2: The authors clarified all the requested points.

Reviewer #3: Overall, the work is carefully carried out and clearly described. It represents a useful systematic study that increases knowledge of lectin-glycan binding features laying the basis to obtain further insight into the field. The minor edits requested have been satisfied. Based on this, I suggest the publication of the paper.

**Have the authors made all data and (if applicable) computational code underlying the findings in their manuscript fully available?**

Reviewer #1: Yes

Reviewer #2: Yes

Reviewer #3: Yes

PLOS authors have the option to publish the peer review history of their article (what does this mean?). If published, this will include your full peer review and any attached files.

Reviewer #1: **Yes: **elisa fadda

Reviewer #2: No

Reviewer #3: **Yes: **Giulia Paiardi

Figure Files:

Data Requirements:

Reproducibility:

References:

---

## [Editor Report · Decision Letter 2]

22 Sep 2021

Dear Dr. Bailey-Kellogg,

We are pleased to inform you that your manuscript 'Comprehensive analysis of lectin-glycan interactions reveals determinants of lectin specificity' has been provisionally accepted for publication in PLOS Computational Biology.

Best regards,

Rebecca C. Wade

Associate Editor

PLOS Computational Biology

Arne Elofsson

Deputy Editor

PLOS Computational Biology

---

## [Editor Report · Acceptance letter]

1 Oct 2021

PCOMPBIOL-D-21-01090R2

Comprehensive analysis of lectin-glycan interactions reveals determinants of lectin specificity

Dear Dr Bailey-Kellogg,

I am pleased to inform you that your manuscript has been formally accepted for publication in PLOS Computational Biology. Your manuscript is now with our production department and you will be notified of the publication date in due course.

With kind regards,

Andrea Szabo
